# Molecular basis of P[II] major human rotavirus VP8* domain recognition of histo-blood group antigens

**Shenyuan Xu**[1], **Luay U. Ahmed** [1], **Michael Robert Stuckert** [1], **Kristen Rose McGinnis**[1], **Yang Liu**[2,3], **Ming Tan**[2,4], **Pengwei Huang** [2], **Weiming Zhong**[2], **Dandan Zhao**[2], **Xi Jiang**[2,4]*, **Michael A. Kennedy** [1]*

**1** Department of Chemistry and Biochemistry, Miami University, Oxford, Ohio, United States of America, **2** Division of Infectious Diseases, Cincinnati Children's Hospital Medical Center, Cincinnati, Ohio, United States of America, **3** Tianjin Key Laboratory of Molecular Nuclear Medicine, Institute of Radiation Medicine, Chinese Academy of Medical Sciences and Peking Union Medical College, Tianjin, China, **4** University of Cincinnati College of Medicine, Cincinnati, Ohio, United States of America

* Jason.Jiang@cchmc.org (XJ); kennedm4@miamioh.edu (MAK)

**Data Availability Statement:** All structural data files are available from the PDB database (accession numbers, 6NIW and 6OAI). All NMR

## Abstract

Initial cell attachment of rotavirus (RV) to specific cell surface glycan receptors, which is the essential first step in RV infection, is mediated by the VP8* domain of the spike protein VP4. Recently, human histo-blood group antigens (HBGAs) have been identified as receptors or attachment factors for human RV strains. RV strains in the P[4] and P[8] genotypes of the P [II] genogroup share common recognition of the Lewis b (Le$^b$) and H type 1 antigens, however, the molecular basis of receptor recognition by the major human P[8] RVs remains unknown due to lack of experimental structural information. Here, we used nuclear magnetic resonance (NMR) spectroscopy-based titration experiments and NMR-derived high ambiguity driven docking (HADDOCK) methods to elucidate the molecular basis for P[8] VP8* recognition of the Le$^b$ (LNDFH I) and type 1 HBGAs. We also used X-ray crystallography to determine the molecular details underlying P[6] recognition of H type 1 HBGAs. Unlike P[6]/P[19] VP8*s that recognize H type 1 HBGAs in a binding surface composed of an α-helix and a β-sheet, referred as the "βα binding site", the P[8] and P[4] VP8*s bind Le$^b$ HBGAs in a previously undescribed pocket formed by the edges of two β-sheets, referred to as the "ββ binding site". Importantly, the P[8] and P[4] VP8*s retain binding capability to non-Le$^b$ type 1 HBGAs using the βα binding site. The presence of two distinct binding sites for Le$^b$ and non-Le$^b$ HBGA glycans in the P[8] and P[4] VP8* domains suggests host-pathogen co-evolution under structural and functional adaptation of RV pathogens to host glycan polymorphisms. Assessment and understanding of the precise impact of this co-evolutionary process in determining RV host ranges and cross-species RV transmission should facilitate improved RV vaccine development and prediction of future RV strain emergence and epidemics.

data files are available from the BioMagResBank database (accession numbers, 27591 and 27592).

**Funding:** The author(s) received no specific funding for this work.

**Competing interests:** The authors have declared that no competing interests exist.

## Author summary

Rotaviruses (RV)s are the main cause of severe diarrhea in humans and animals. Significant advances in understanding RV diversity, evolution and epidemiology have been made after discovering that RVs recognize histo-blood group antigens (HBGAs) as host cell receptors or attachment factors. While different RV strains are known to have distinct binding preferences for HBGA receptor ligands, their molecular basis in controlling strain-specific host ranges remains unclear. In this study, we used solution nuclear magnetic resonance spectroscopy and X-ray crystallography to determine the molecular-level details for interactions of the human P[8] and P[6] RV VP8* domains with their HBGA receptors ligands. The distinct binding patterns observed between these major human RVs and their respective glycan ligands provide insight into the evolutionary relationships between different P[II] genotypes that ultimately determine host ranges, disease burden, zoonosis and epidemiology, which may impact future strategies for development of vaccines to protect against RV infections.

## Introduction

Rotaviruses (RVs) cause severe dehydrating gastroenteritis in children younger than five years of age, accounting for two million childhood hospital admissions and leading to approximately 200,000 deaths each year [1–4]. RVs, members of the family *Reoviridae*, are double-stranded RNA viruses that contain a segmented genome, encoding six structural and six nonstructural proteins. The viral genome is encapsulated by three concentric protein layers, with the inner layer made by viral protein 2 (VP2), the intermediate shell by VP6, and the outer layer by glycosylated VP7 [5]. The protease-sensitive VP4 proteins protrude from the outer capsid shell, forming sixty spikes that are responsible for host receptor interactions. The two outer capsid proteins, VP7 and VP4, are determinants of RV G and P types, respectively, and induce G and P type-specific neutralizing antibody responses. Thus, VP7 and VP4 are used to classify RV types, which is the basis of the RV dual-nomenclature system [6]. RVs are genetically diverse and at least 51 P genotypes have been identified to date among human and animal rotaviruses [7]. Previously, the diverse P genotypes among group A RVs have been classified into five genogroups (P[I]-P[V]) based on the VP4 sequences [8–10]. These RVs can infect humans and/or different animal species, among which P[8], P[4], and P[6] RVs in P[II] genogroup are the predominant genotypes causing diseases in humans worldwide [11–13].

 RVs recognize cell surface carbohydrates through the VP4 spike proteins in a process that leads to viral attachment to the host cells in a step that is critical to initiation of host infections. Each VP4 protein is cleaved by trypsin into two domains, VP5* and VP8*, corresponding to the stalk and the distal head of the spike protein, respectively [14]. The distal VP8* head interacts with RV host glycan receptors for viral attachment, while the VP5* stalk assists in viral penetration into host cells [14, 15]. A growing body of data indicate that different P-type RVs exhibit distinct glycan binding specificities that may be responsible for determining host ranges as well as zoonotic transmission. While the sialidase sensitive P[1], P[2], P[3], and P[7] genotypes in the P[I] genogroup recognize sialoglycans for attachment [16–18], many sialidase insensitive RVs in the five genogroups recognize polymorphic histo-blood group antigens (HBGAs) as attachment factors [10, 19–22]. HBGAs are complex carbohydrates that are present on red blood cells and epithelia of the gastrointestinal, respiratory and genitourinary tracts and in exocrine secretions [23, 24]. HBGAs are synthesized by sequential addition of a carbohydrate moiety to a precursor disaccharide in successive steps that are genetically controlled

by ABO (*ABO*, 9q34.1), H (*FUT1*, 19q13.3), secretor (*FUT2*, 19q13.3) and Lewis (*FUT3*, 19p13.3) gene families [25]. For example, the *FUT2* gene encodes an α-1,2-fucosyltransferase that catalyzes addition of an α-1,2-fucose to the precursor oligosaccharides, forming H-type antigens, while the *FUT3* gene encodes an α-1,3/4-fucosyltransferase that turns precursor oligosaccharides or H-type antigens into Lewis a (Le$^a$) or Lewis b (Le$^b$) antigens.

Increasing evidence shows a connection between HBGA phenotypes and RV epidemiology, indicating that HBGAs play an important role in determining interspecies transmission and disease burden of RVs. For example, whereas P[6] and P[19] RVs in the P[II] genogroup are reported to only recognize H type 1 antigen in the absence of the Lewis fucose, P[4] and P[8] RVs bind both Lewis epitope modified (Le$^b$) and unmodified H type 1 antigens [20, 26], supporting the observation that the Lewis negative individuals may have increased risk of P[6] RV infection whereas P[4] and P[8] RVs have broader host ranges to Lewis and secretor positive individuals [27, 28]. The P[9], P[14], and P[25] genotypes in the P[III] genogroup recognize type A HBGAs [10], which explains that P[14] infection in both humans and animals may be due to type A HBGAs that are shared between human and animal species [10, 19, 29]. The P[11] RVs in the P[IV] genogroup recognize type 1 and type 2 HBGA precursors [21, 22, 30]. Previous studies demonstrate that bovine P[11] VP8* only recognizes type 2 HBGAs whereas human neonatal P[11] VP8* binds both type 1 and type 2 HBGAs, which is consistent with the finding that abundant type 2 glycans were found in bovine milk while both type 1 and type 2 glycans were found in human milk [30–32].

To further understand the role that HBGAs play in determining RV host ranges, evolution, and zoonosis, it is necessary to characterize the molecular basis of the interactions between VP8* domains and their glycan/HBGA attachment factors. The RV VP8* domain adopts a classical galectin-like fold with the central structure being an anti-parallel β-sandwich formed by a five-stranded β-sheet and a six-stranded β-sheet. Past crystallography studies have demonstrated that the VP8* domain of P[3] RVs binds sialic acids in an open-ended, shallow groove between the two β-sheets. The same binding site is used by P[7] and P[14] RVs to bind mono-sialodihexosylganglioside GM3 and A-type HBGAs, respectively [19, 33, 34]. The binding cleft is wider in P[11] than in P[3]/P[7] and P[14] strains and the binding site is shifted to span almost the entire length of the cleft between the two β-sheets. A significant advance in understanding the molecular basis of HBGA-VP8* interactions was achieved through nuclear magnetic resonance (NMR) spectroscopy and crystallographic studies of P[19] binding to type 1 HBGA, which led to the discovery of a completely new glycan binding site composed of the N-terminal α-helix and the β-sheet containing βB, βI, βJ, βK stands [26, 35]. Recently, P[4] and P[6] VP8* were found to interact with H-type 1 HBGA using the same binding pocket formed by the β-sheet and N-terminal α-helix [36].

The crystal structure of an apo form of P[8] human RV (Wa-like) has been reported previously [37], and the crystal structures of P[8] in complex with the type 1 precursor and the H type 1 trisaccharide, as well as the P[8] in complex with LNFP I, have been reported recently [38, 39]. However, structural data regarding P[8] recognition of Lewis b antigen is not currently available. As an alternative to X-ray crystallography, NMR experiments can be used to provide valuable information about the protein ligand interactions at the atomic level [40–42]. In this study, we employed high ambiguity driven docking (HADDOCK) [42, 43] driven by NMR chemical shift perturbation data to elucidate the molecular basis for glycan recognition by the VP8* domain of the P[8] and P[6] major human RVs. We found that P[6] VP8* bound the tetra- (LNT) and penta- (LNFP I) saccharide of type 1 HBGAs without Lewis epitopes in an open pocket formed by an α-helix and β-sheet consisting of βK, βJ, βI, βB strands, consistent with the crystal structure of LNFP I-bound P[6]. However, P[8] VP8* bound the Le$^b$ tetrasaccharide and a Le$^b$-containing hexa-saccharide (LNDFH I), in a shallow cleft formed by the

two twisted antiparallel β-sheets. The discovery and documentation of the complex and variable attachment factor binding patterns of the four P[II] RVs (P[4], P[6], P[8] and P[19]) reported here and elsewhere continues to expand our understanding of the evolution, host ranges, disease burden and zoonosis of P[II] RVs.

## Results

### Identification of specific HBGA glycan interactions with P[8] VP8*

P[8] VP8* has been reported to bind the H-type 1 trisaccharide, the Le$^b$ tetra-saccharide and the hexa-saccharide LNDFH I that contains the Lewis epitope (Fig 1A) [26]. Infection assays conducted in this study using these glycans revealed specific blocking activities against P[8] Wa strain replication in HT29 cells (Fig 1E), however structures of their complexes do not exist at this time. In order to investigate the structural basis for interactions between the P[8] VP8* and these glycans, proton saturation transfer difference (STD) NMR experiments were used to characterize the specific interactions between the P[8] RV VP8* domain and the Le$^b$ tetra-saccharide and hexa-saccharide LNDFH I HBGAs. Proton NMR resonances unique to the protein were selectively saturated and saturation transfer to protons on the bound ligand measured using STD NMR experiments. Protons of glycan sugar moieties that displayed efficient saturation transfer, i.e. large STD amplification factors, were inferred to be in direct contact with the VP8* surface, whereas protons of glycan sugar moieties free from saturation transfer were assumed to be pointing away from the protein surface [44, 45]. Complete glycan proton chemical shift assignments enabled determination of HBGA saccharide moieties in direct contact with the protein (S1 Fig). P[8] VP8* was found to bind both the Le$^b$ tetra-saccharide (Fucα1-2Galβ1–3[Fucα1–4]GlcNAc) and LNDFH I (Fucα1-2Galβ1–3[Fucα1–4]GlcNAcβ1-3Galβ1-4Glc) (Fig 2A and Fig 2C). Protons in the type 1 precursor Galβ1-3GlcNAc motif of both the Le$^b$ tetra-saccharide (Fig 2B) and LNDFH I (Fig 2D) experienced strong magnetization saturation transfer, indicating that the type 1 precursor Galβ1-3GlcNAc motif was important for P[8] VP8* recognition in both HBGA glycans. The secretor fucose and Lewis fucose moieties of the Le$^b$ tetra-saccharide and LNDFH I were also involved in binding interactions (Fig 2), however, the secretor fucose in both Le$^b$ tetra-saccharide and LNDFH I, and Lewis fucose in LNDFH I received relatively weak saturation transfer, while the Lewis epitope in Le$^b$ tetra-saccharide experienced relatively strong magnetization transfer. The galactose (Gal-IV) and glucose (Glc-V) moieties of LNDFH I also appeared to make direct contact with the P[8] VP8* surface, however their protons experienced relatively weak saturation transfer similar to that observed for the secretor and Lewis fucose moieties in LNDFH I. Analysis of the NMR titration experiments indicated that P[8] VP8* bound more tightly to LNDFH I with a dissociation constant ($K_d$) of 6.3 ± 0.8 mM, likely due to the additional interactions with the Gal-IV and Glc-V sugar moieties, compared to Le$^b$ tetra-saccharide, which bound more weakly with a $K_d$ value of 20.3 ± 4.8 mM (S4 Fig).

### Identification of specific HBGA glycan interactions with P[6] VP8*

P[6] VP8* has been previously reported to bind the non-Le$^b$ LNT and LNFP I glycans (Fig 1C) [26], and the crystal structure of the P[6] VP8* domain complex with LNFP I has been reported [36], however its structure with LNT has not been reported at this time. Here, proton STD NMR experiments were used to characterize the interactions between the P[6] RV VP8* domain and the LNT and LNFP I HBGAs (Fig 3). STD NMR experiments indicated that P[6] VP8* bound both LNT (Galβ1-3GlcNAcβ1-3Galβ1-4Glc) and LNFP I (Fucα1-2Galβ1-3GlcNAcβ1-3Galβ1-4Glc) glycans (Fig 3). Protons in the Galβ1-3GlcNAcβ1-3Gal chain of both LNT and LNFP I experienced strong saturation transfer indicating that this motif,

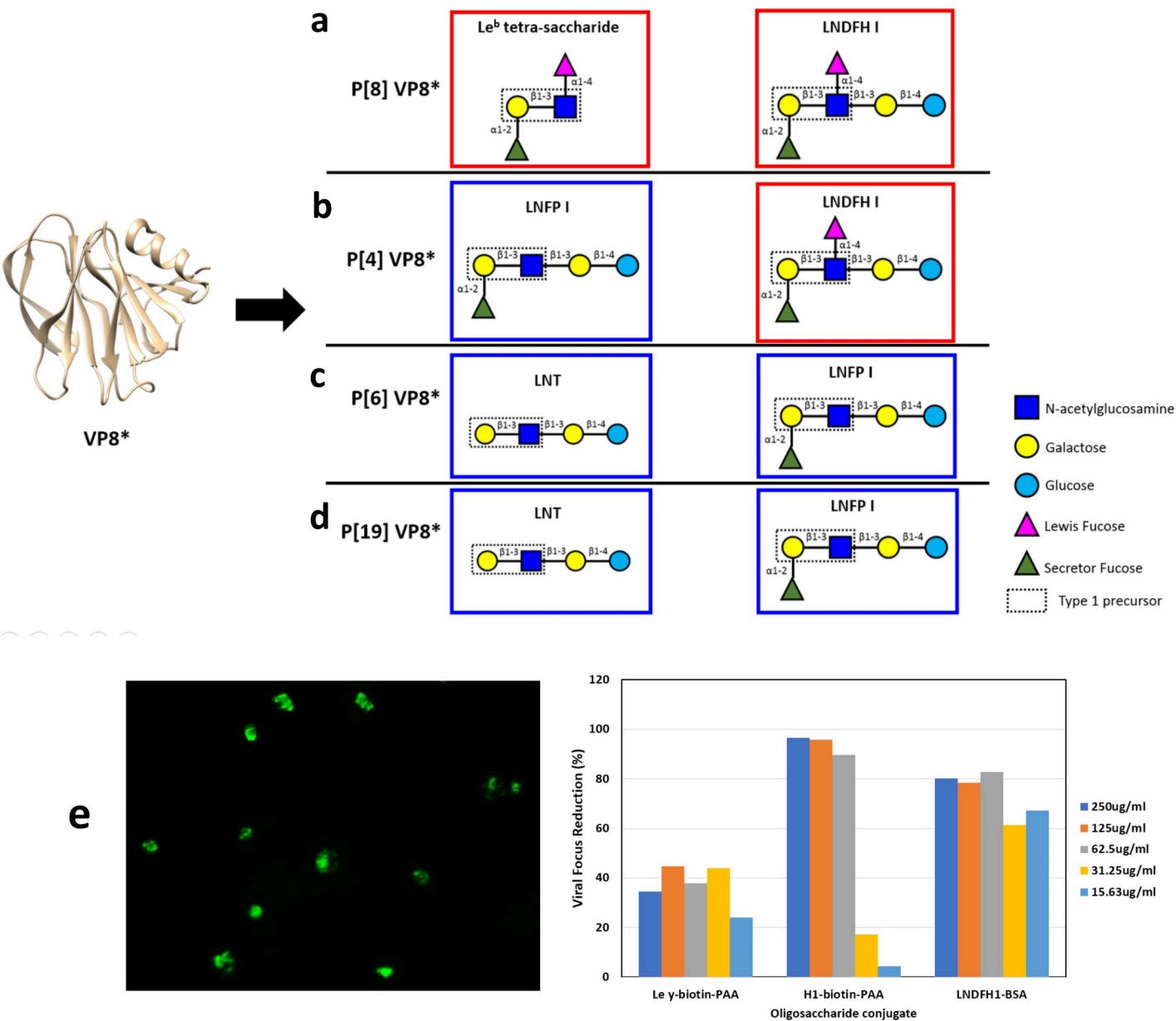

**Fig 1. Schematic representation of VP8* from P[II] genogroup with its corresponding binding glycan structures.** (a) Schematic representation of P[8] VP8* and its binding glycan structures, Le^b tetra-saccharide and LNDFH I. (b) Schematic representation of P[4] VP8* and its binding glycan structures, LNDFH I and LNFP I. (c) Schematic representation of P[6] VP8* and its binding glycan structures, LNT and LNFP I. (d) Schematic representation of P[19] VP8* and its binding glycan structures, LNT and LNFP I. The red boxes encircle Le^b glycans and the blue boxes encircle non Le^b glycans. (e) Inhibition of rotavirus Wa strain replication in HT29 cells. Left) Representative indirect immunofluorescence assay (IFA) microscopy image of P[8] RV Wa strain replication focuses in HT29 cells. Right) Quantitation of viral replication focus reductions observed in wells incubated with LNDFH1-BSA and H type 1-biotin-PAA in compared with cell culture wells with media only. Le^y-PAA is a negative ligand control.

common to both LNT and LNFP 1, was important in P[6] VP8*-HBGA interactions. The terminal glucose of LNT and LNFP I also experienced saturation transfer from the protein, although the transfer was unevenly distributed in the glucose between the two glycans, with H1 in the Glc-V moiety of LNFP I having stronger saturation transfer. Fuc-I in LNFP I appeared to point away from the protein surface as its protons received only weak saturation

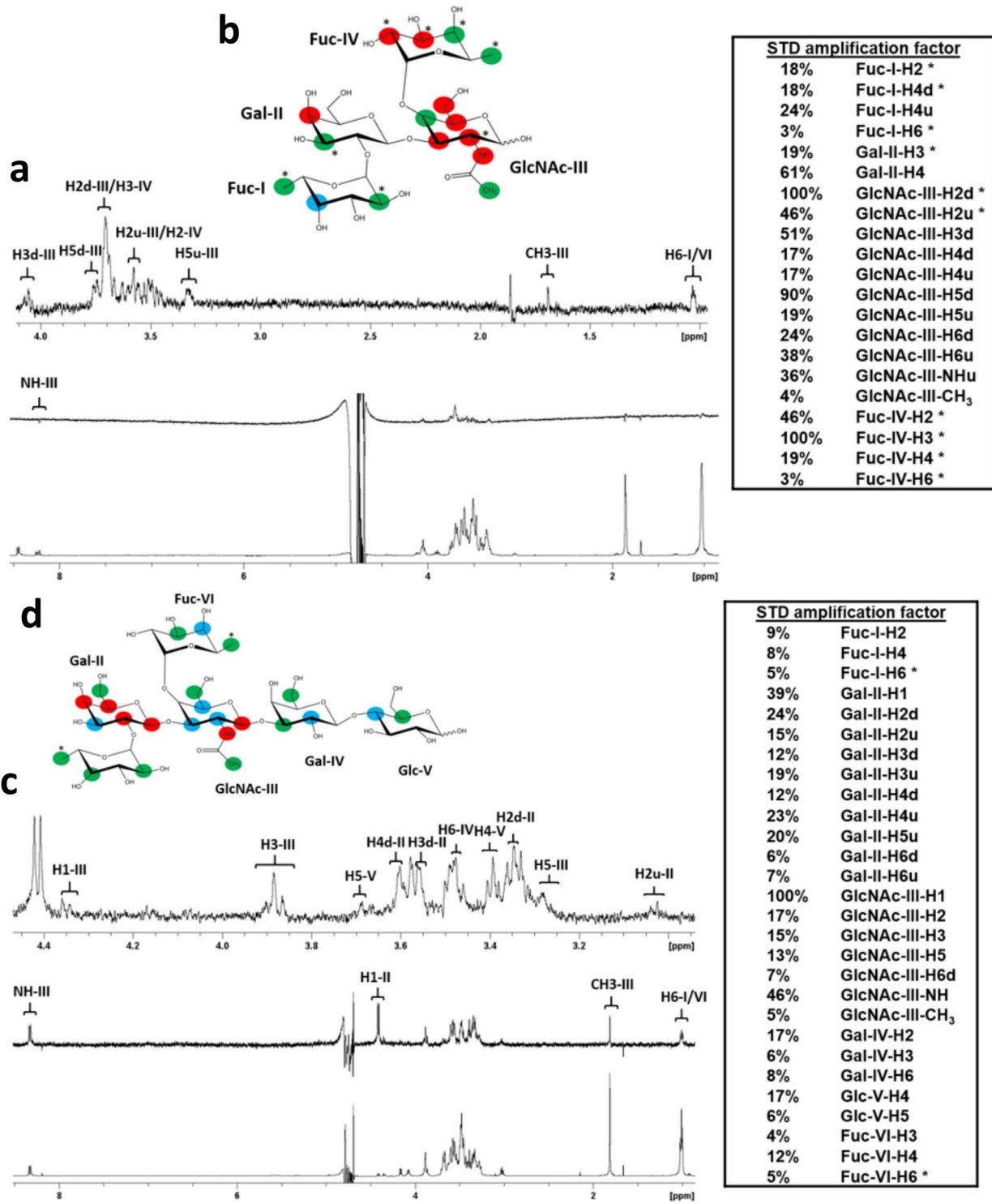

| STD amplification factor | |
| --- | --- |
| 18% | Fuc-I-H2 * |
| 18% | Fuc-I-H4d * |
| 24% | Fuc-I-H4u |
| 3% | Fuc-I-H6 * |
| 19% | Gal-II-H3 * |
| 61% | Gal-II-H4 |
| 100% | GlcNAc-III-H2d * |
| 46% | GlcNAc-III-H2u * |
| 51% | GlcNAc-III-H3d |
| 17% | GlcNAc-III-H4d |
| 17% | GlcNAc-III-H4u |
| 90% | GlcNAc-III-H5d |
| 19% | GlcNAc-III-H5u |
| 24% | GlcNAc-III-H6d |
| 38% | GlcNAc-III-H6u |
| 36% | GlcNAc-III-NHu |
| 4% | GlcNAc-III-CH3 |
| 46% | Fuc-IV-H2 * |
| 100% | Fuc-IV-H3 * |
| 19% | Fuc-IV-H4 * |
| 3% | Fuc-IV-H6 * |

| STD amplification factor | |
| --- | --- |
| 9% | Fuc-I-H2 |
| 8% | Fuc-I-H4 |
| 5% | Fuc-I-H6 * |
| 39% | Gal-II-H1 |
| 24% | Gal-II-H2d |
| 15% | Gal-II-H2u |
| 12% | Gal-II-H3d |
| 19% | Gal-II-H3u |
| 12% | Gal-II-H4d |
| 23% | Gal-II-H4u |
| 20% | Gal-II-H5u |
| 6% | Gal-II-H6d |
| 7% | Gal-II-H6u |
| 100% | GlcNAc-III-H1 |
| 17% | GlcNAc-III-H2 |
| 15% | GlcNAc-III-H3 |
| 13% | GlcNAc-III-H5 |
| 7% | GlcNAc-III-H6d |
| 46% | GlcNAc-III-NH |
| 5% | GlcNAc-III-CH3 |
| 17% | Gal-IV-H2 |
| 6% | Gal-IV-H3 |
| 8% | Gal-IV-H6 |
| 17% | Glc-V-H4 |
| 6% | Glc-V-H5 |
| 4% | Fuc-VI-H3 |
| 12% | Fuc-VI-H4 |
| 5% | Fuc-VI-H6 * |

**Fig 2. STD NMR analysis of Le^b tetrasaccharide and LNDFH I glycan interactions with the P[8] VP8* domain.** (a) NMR spectra of Le^b tetra-saccharide in complex with P[8] VP8*. The bottom spectrum in (a) is ^1H NMR reference spectrum of P[8] VP8* (33 μM) with Le^b tetra-saccharide (1.86 mM). The middle spectrum in (a) is STD NMR spectrum of P[8] VP8* (33 μM) with Le^b tetra-saccharide (1.86 mM). The protein was saturated with a cascade of 40 Gaussian-shaped pulses at 0.11 ppm, and the off-resonance was set to 50 ppm. The upper spectrum in (a) is the expansion of the STD NMR from 4.1 to 0.9 ppm. (b) The epitope mapping of Le^b tetra-saccharide when bound to P[8] based on the STD effects: red, strong STD NMR effects (>30%); blue, medium STD NMR effects (20%-30%); green, weak STD NMR effects (<20%). (c) NMR spectra pf LNDFH I in complex with P[8] VP8*. The bottom spectrum in (c) is ^1H NMR reference spectrum of P[8] VP8* (46 μM) with LNDFHI (2.3 mM). The middle spectrum in (c) is STD NMR spectrum of P[8] VP8* (46 μM) with LNDFHI (2.3 mM). The protein was saturated with a

cascade of 40 Gaussian-shaped pulses at -0.25 ppm, and the off-resonance was set to 50 ppm. The upper spectrum in (c) is the expansion of the STD NMR from 4.5 to 3.0 ppm. (d) The epitope mapping of LNDFH I when bound to P[8] based on the STD effects: red, strong STD NMR effects (>20%); blue, medium STD NMR effects (10%-20%); green, weak STD NMR effects (<10%). "*" means the overlapping of signals in the STD NMR spectrum. "d" indicates a downfield proton, "u" indicates an upfield proton.

transfer. The affinity of P[6] VP8* to LNT was higher ($K_d$ = 2.5 ± 0.2 mM) compared to that of LNFP I ($K_d$ = 13.6 ± 1.7 mM), based on the analysis of the NMR titration data (S6 Fig).

## Mapping HBGA glycan interactions onto the surface of P[8] VP8* using chemical shift perturbations observed in HSQC NMR titration experiments

The P[8] VP8* backbone resonances were assigned using triple resonance NMR experiments and the backbone chemical shifts were deposited to the BioMagResBank database (Entry ID: 27592). The 2D $^1$H-$^{15}$N HSQC spectrum of P[8] VP8* was of excellent quality (S2A Fig) and the backbone assignments for NH pairs were 94% complete. Titrations of Le$^b$ tetra-saccharide and LNDFH I into solutions of $^{15}$N-labeled P[8] VP8* caused significant chemical shift perturbations and disappearance of some resonances in the $^1$H-$^{15}$N-HSQC spectra, while the chemical shifts of the majority signals were slightly or not affected (Fig 4A and 4D). Complete profiles of chemical shift perturbations of P[8] VP8* upon addition of Le$^b$ tetra-saccharide and LNDFH I are shown in (S3 Fig). Titration with either Le$^b$ tetra-saccharide or LNDFH I resulted in a common pattern of affected residues, with L157 of βH strand, K168 of βI strand, G178, E179, and A183 of βJ-K loop, and D186 of βK strand exhibiting large chemical shifts changes (Fig 4). In addition to the commonly-affected residues, the two glycans also caused some slightly different chemical shift perturbations, both on the periphery and outside the common area. For example, addition of the Le$^b$ tetra-saccharide caused large chemical shifts changes clustered near the common area, including T156 of βH strand and F176 of βJ strand, and A107 of βD strand, however, G145 of βG-H loop also showed large chemical shifts changes, which was distant from the common area (Fig 4B and 4C). Upon addition of LNDFH I, additional amino acids with significant chemical shifts perturbations that clustered near the common area included R154 of βH strand, H177 of βJ strand, T185 of βK strand, and several additional residues that were not clustered near the common area experienced large chemical shifts perturbations, including T78 and D79 of βA-B loop, T161 of βH-I loop, V164 and G165 of βI strand, I207 of βM strand (Fig 4E and 4F).

## Mapping HBGA glycan interactions onto the surface of P[6] VP8* using chemical shift perturbations observed in HSQC NMR titration experiments

The P[6] VP8* backbone resonances were assigned using triple resonance experiments and the backbone chemical shifts were deposited to the BioMagResBank database (Entry ID: 27591). The 2D $^1$H-$^{15}$N HSQC spectrum of the P[6] VP8* domain was of excellent quality (S2B Fig) and the backbone assignments for NH pairs were 92% complete. Titration of $^{15}$N-labeled P[6] VP8* with LNT and LNFP I caused some resonances to experience significant chemical shift perturbations or to disappear altogether, while the remaining peaks were weakly or not affected (Fig 5A and 5D, and S5 Fig). Addition of LNT caused significant chemical shifts perturbations clustered in the region composed of the following residues: N78 of the βA-B loop, Y80 of βB strand, F169 of βI strand, W174 of βJ strand, T184 and T185 of βK strand, Q211, E212, K214, C215, S216 of αA helix (Fig 5B and 5C). In addition, E141 of βG strand, T190, S191, and N192 of βK-L loop, Q89 of βB-C loop, H201 of βL-M loop, which were not in the main clustered region, also showed large chemical shifts changes. Upon adding LNFP I, residues that showed significant changes of chemical shifts were clustered in the area that

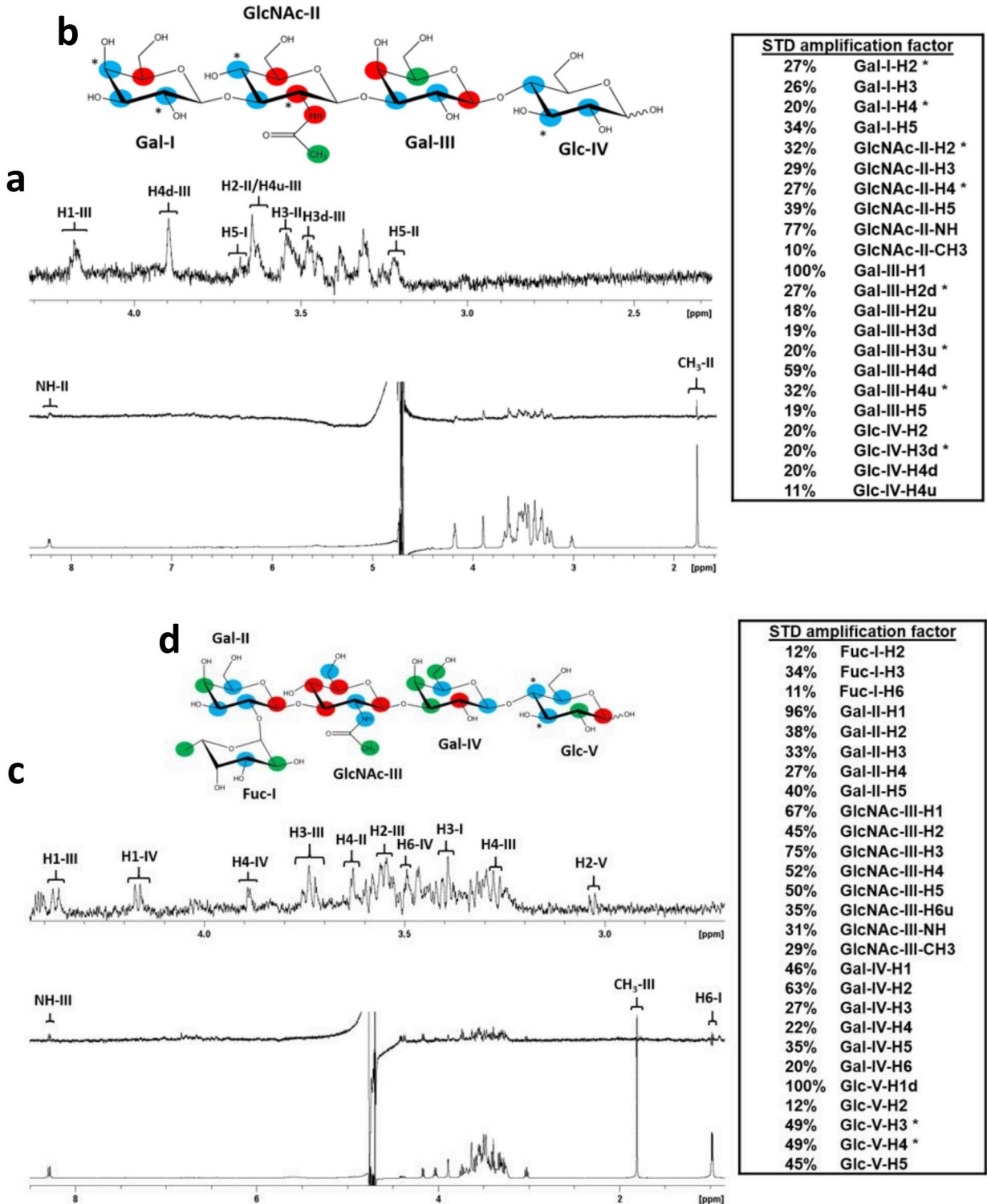

**Fig 3. STD NMR analysis of LNT and LNFPI glycan interactions with the P[6] VP8* domain.** (a) NMR spectra pf LNT in complex with P[6] VP8*. The bottom spectrum in (a) is the ${}^1$H NMR reference spectrum of P[6] VP8* (46 μM) with LNT (4.6 mM).The middle spectrum in (a) is the STD NMR spectrum of P[6] VP8* (46 μM) with LNT (4.6 mM). The protein was saturated with a cascade of 40 Gaussian-shaped pulses at 0.63 ppm, and the off-resonance was set to 50 ppm. The upper spectrum in (a) is the expansion of the STD NMR from 4.3 to 2.3 ppm. (b) The epitope map of LNT when bound to P[6] based on the STD effects: red, strong STD NMR effects (>30%); blue, medium STD NMR effects (20%-30%); green, weak STD NMR effects (<20%). (c) NMR spectra pf LNFPI in complex with P[6] VP8*. The bottom spectrum in (c) is the ${}^1$H NMR

reference spectrum of P[6] VP8* (46 μM) with LNFPI (2.3 mM). The middle spectrum in (c) is the STD NMR spectrum of P[6] VP8* (46 μM) with LNFPI (2.3 mM). The protein was saturated with a cascade of 40 Gaussian-shaped pulses at -0.21 ppm, and the off-resonance was set to 50 ppm. The upper spectrum is the expansion of the STD NMR from 4.4 to 1.0 ppm. (d) The epitope map of LNFPI when bound to P[6] based on the STD effects: red, strong STD NMR effects (>50%); blue, medium STD NMR effects (30%-50%); green, weak STD NMR effects (<30%). "*" means the overlap of signals in the STD NMR spectrum. "d" indicates a downfield proton, "u" indicates an upfield proton.

consisted of N78 of βA-B loop, Y80 of βB strand, F169 of βI strand, V173 and W174 of βJ strand, T184, T185, D186, and S188 of βK strand, and Q211, E212, K214, C215, S216 of αA helix (Fig 5E and 5F). In addition, T190, S191, N192, and L193 of βK-L loop had large chemical shifts perturbation upon adding LNFP I.

## Mapping HBGA glycan interactions onto the surface of P[8] VP8* mutants using chemical shift perturbations observed in HSQC NMR titration experiments

To confirm residues essential to P[8] VP8* in recognizing the Le[b]-containing type 1 HBGAs, we prepared six P[8] mutants based on titration results of native P[8] VP8* with LNDFH I including three amino acids that occurred in the common area, i.e. the putative binding site (R154, H177, D186) and three amino acids that experienced significant chemical shifts but were distal from the putative binding site (T78, D79, K168). When LNDFH I was titrated into the T78A-P[8], D79A-P[8], and K168A-P[8] mutants distal from the putative binding site, the mutants retained binding to the glycan whereas mutations in the putative binding site, i.e. the R154A-P[8], H177A-P[8], and D186A-P[8] mutants, exhibited a significant loss of binding to the glycan (S7 Fig). In the T78A-P[8] mutant, R154, T156, L157 of βH, V164, G165, K168 of βI strand, H177 of βJ strand, G178, E179, and A183 of βI-J loop, and T184, T185, D186 of βK strand experienced large chemical shifts perturbations (Fig 6A, 6B and 6C). In the D79A-P[8] mutant, R154, R155, T156, and L157 of βH strand, V164, G165 of βI strand, H177 of βJ strand, G178, E179, A183 of βI-J loop, and T184, T185, D186, S187 of βK strand demonstrated significant chemical shift perturbations (Fig 6D, 6E and 6F). In K168A-P[8], R154, and L157 of βH strand, V164 and G165 of βI strand, G178, E179, and A183 of βI-J loop, T185 and D186 of βK strand showed large chemical shift perturbations (Fig 6G, 6H and 6I). It was deduced from all the titration experiments that P[8] uses a pocket mainly formed by the edges of the βH, βI, βJ, and βK strands to recognize the LNDFH I.

## Models of the structures of the P[8] VP8* domain bound to the glycan ligands

Superposition of the top five best-scoring P[8] VP8* structures from HADDOCK docking showed that the bound structures of the Le[b] tetra-saccharide and LNDFH I clustered in the same shallow cleft formed by the edges of two β-sheets (referred to hereafter as the "ββ binding site") with the only difference being that LNDFH I covered more surface area (Fig 7). The ββ binding site was mainly composed of the βH strand, the C-terminus part of βJ strand, the N-terminus part of βK strand, and the βJ-K loop connecting the two strands.

## Models of the structures of the P[6] VP8* domain bound to the glycan ligands

The top five best-scoring P[6] VP8* structures revealed that LNT and LNFP I clustered in a shallow groove composed of the N-terminal α-helix and the β-sheet composed of the βK, βJ, βI, βB strands (hereafter referred to as the "βα binding site") (S8 Fig). The binding interface

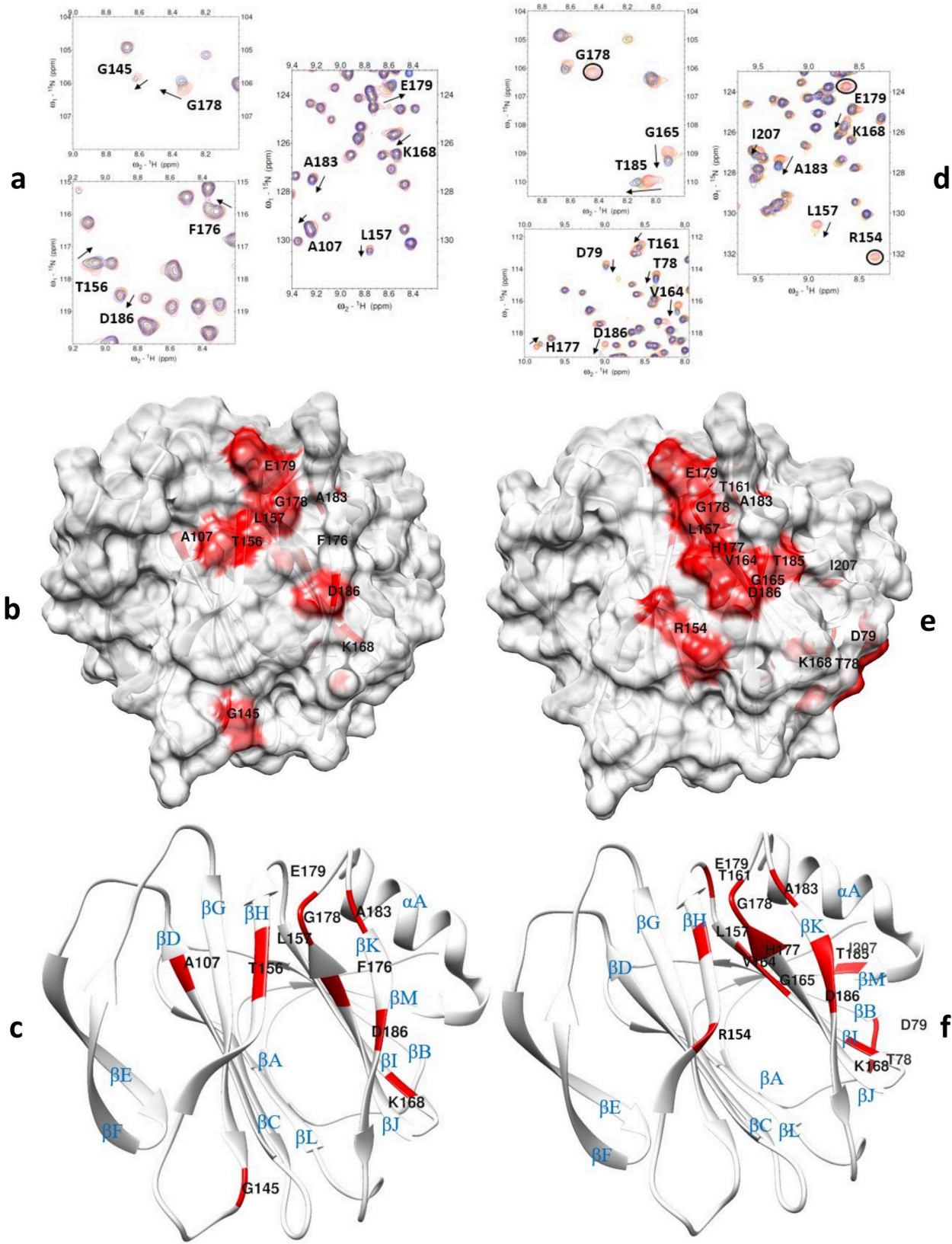

**Fig 4. NMR titration analysis of P[8] VP8* domain with Le^b and LNDFH I glycans.** (a) Chemical shift changes in P[8] VP8* [PDB ID: 2DWR] upon addition of Le^b tetra-saccharide. The NMR data correspond to increasing ligand/protein ratios of 0:1 (red), 8:1 (orange), 18:1 (green), and 30:1 (blue). (b) Location of the large chemical shift changes on the P[8] VP8* surface [PDB ID: 2DWR] upon binding to Le^b tetra-saccharide. Amino acids with chemical shift changes greater than 2.5σ or disappeared after titration are colored with red. (c) Ribbon Diagram shows the location of large chemical shift changes on the P[8] VP8* surface upon binding to Le^b tetra-saccharide. The secondary structure is labeled. (d) Chemical shift changes in P[8] VP8* upon addition of LNDFH I. The NMR data correspond to increasing ligand/protein ratios of 0:1 (red), 6:1 (orange), 18:1 (green), and 25:1 (blue). The arrows show residues that have chemical shifts changes upon titration. Peaks that disappeared upon titration are circled. (e): Location of the large chemical shift changes on the P[8] VP8* surface [PDB ID: 2DWR] upon binding to LNDFH I. Amino acids with chemical shift changes greater than 1.5σ or disappeared after titration are colored with red. (f): Ribbon Diagram shows the location of large chemical shift changes on the P[8] VP8* surface upon binding to LNDFH I. The secondary structure is labeled.

identified by the HADDOCK docking agreed with the binding region revealed by the X-ray crystal structure of LNFP I-bound P[6] (strain BM11596).

## Crystal structure of the P[6] VP8* domain in complex with LNFP I

The crystal structures of P[6] (strain BM11596) and P[6] (strain BM11596) in complex with LNFP I were solved (Table 1). The VP8* of P[6] BM11596 strain adopted a classical galectin-like fold as similar to other known VP8* structures, with two twisted antiparallel β-sheets consisting of strands A, L, C, D, G, H and M, B, I, J, K, respectively. The specific interactions between LNFP I and P[6] VP8* were revealed by the crystal structure (Fig 8 and S9 Fig). The Galβ1-3GlcNAc type 1 precursor in the LNFP I-bound P[6] VP8* inserted into the pocket formed by W81, M167, A183, T184, T185, R209, E212, and appeared to be stabilized through hydrogen bonding and hydrophobic interactions. The Gal-IV moiety was stabilized by F168, Y170, and W174 through hydrophobic interactions and the terminal Glc-V moiety was stabilized by Y170, S172, Y187 though hydrophobic interactions and N171 through hydrogen bonding. The Fuc-I pointed away from the protein surface, consistent with the STD NMR results.

## Structural conservation and variations within the VP8* glycan binding pocket

When the sequences of different genotypes of VP8* with known structures and experimentally documented binding interfaces were aligned (Fig 9), it was found that RV VP8*s use one or both of two distinct binding pockets to accommodate different glycans (Fig 10), either the ββ binding site or the βα binding site, however, slight sequence and structural variations among different genotypes caused glycans to bind differently within the binding pockets in a genotype-specific manner (Fig 10). P[3]/P[7] in P[I] and P[14] RVs in P[III] use the ββ binding site as a glycan binding interface in which R101 in the βC-D loop and motif 187-XYYX-190 in the βK strand are highly conserved (Fig 9). The conserved pattern observed in P[I] and P[III] genogroups was not found in other genogroups. For example, compared to R101 and Y188 in P[3]/P[7] and P[14], amino acids 101 and 188 are phenylalanine and threonine respectively in the P[11] VP8* in the P[IV] genogroup, whose binding interface covers almost the entire length of βH strand and the C-terminal of βJ strand as well as the βJ-K loop (Fig 10C).

For P[6] and P[19] in the P[II] genogroup, the binding interface is shifted to the βα binding site (Fig 9 and Fig 10). Some key residues involved in the binding pocket in the P[II] genogroup are highly conserved, e.g. W81 in the βB strand, W174 in the βJ strand, T184 and T185 in the βK strand, R209 in the βm-αA loop, and E212 in the αA helix (Fig 9 and Fig 10). These highly conserved key residues in the P[II] genogroup are consistent with the crystal structures showing that P[4]/P[6]/P[8]/P[19] in the P[II] genotype can bind the Lewis negative type 1 HBGAs in the βα site [35, 36, 38, 39] (Fig 10A, 10B, 10C, 10D and 10E). On the other hand,

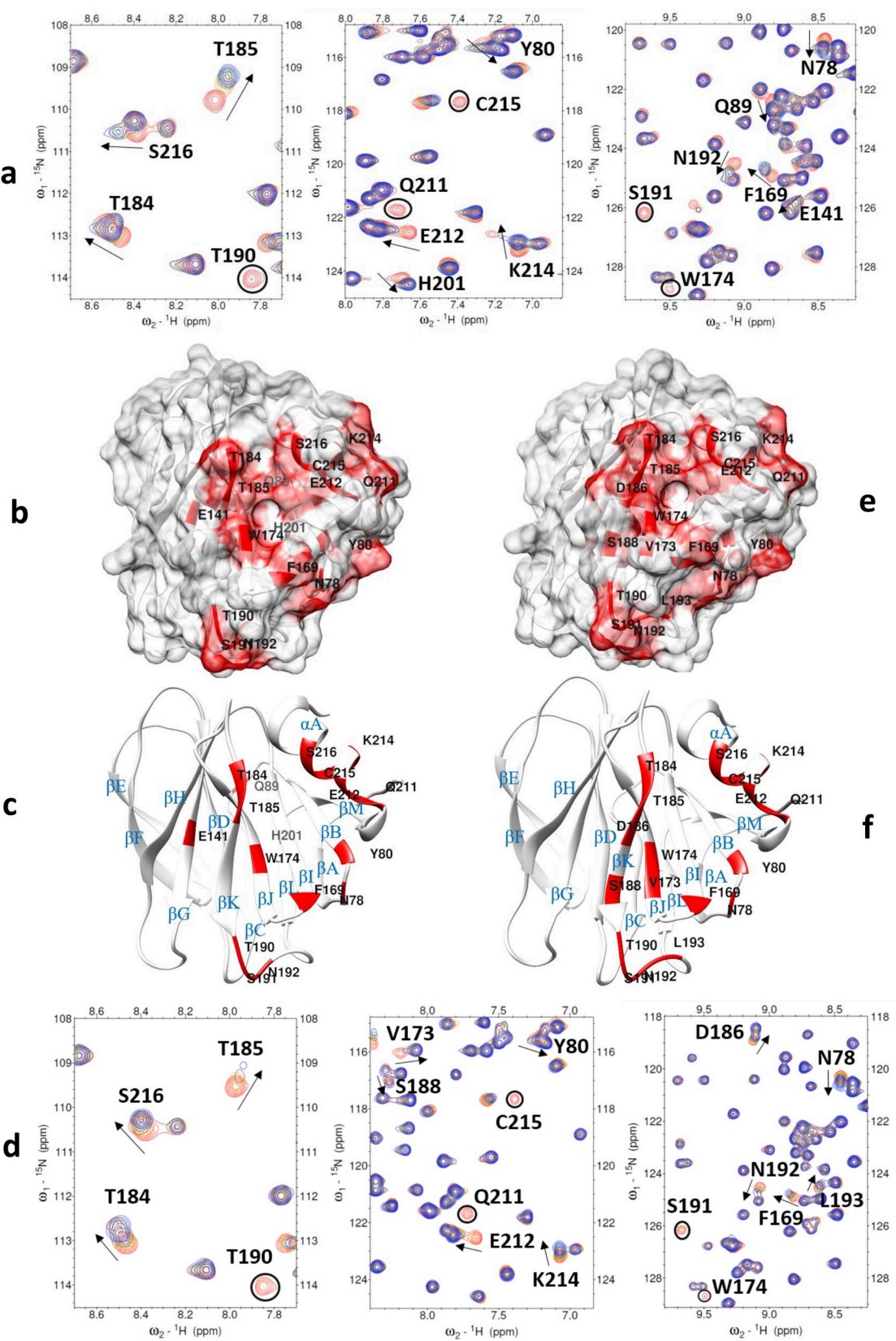

**Fig 5. NMR titration analysis of P[6] VP8\* domain with LNT and LNFP I glycans.** (a) Chemical shift changes in P[6] VP8\* [PDB ID: 6NIW] upon addition of LNT. The NMR data correspond to increasing ligand/protein ratios of 0:1 (red), 8:1 (orange), 18:1 (green), and 25:1 (blue). (b) Location of the large chemical shift changes on the P[6] VP8\* surface [PDB ID: 6NIW] upon binding to LNT. Amino acids with chemical shift changes greater than 2σ or disappeared after titration are colored with red. (c) Ribbon Diagram shows the location of large chemical shift changes on the P[8] VP8\* surface upon binding to Le[b] tetra-saccharide. The secondary structure is labeled. (d) Chemical shift changes in P[6] VP8\* upon addition of LNFP I. The NMR data correspond to increasing ligand/protein ratios of 0:1 (red), 8:1 (orange), 18:1 (green), and 35:1 (blue). The arrows show residues that have chemical shifts changes upon titration. Peaks that disappeared upon titration are circled. (e) Location of the large chemical shift changes on the P[6] VP8\* surface [PDB ID: 6NIW] upon binding to LNFPI. Amino acids with chemical shift changes greater than 1.5σ or disappeared are colored with red. (f) Ribbon Diagram shows the location of large chemical shift changes on the P[6] VP8\* surface upon binding to LNFP I. The secondary structure is labeled.

the key residues of the ββ binding site that enable binding to the Le[b] tetra-saccharide and LNDFH I are highly conserved only between P[4] and P[8] (Fig 9). Thus, we hypothesize that both P[4] and P[8] can bind the Lewis-containing LNDFH I in the ββ binding site and the Lewis negative type 1 HBGAs in the βα binding interfaces while P[6] and P[19] can only bind the latter ones. Because amino acids in the ββ binding site of the BM13851 strain of P[8] are same as those residues in the Wa strain and other strains of P[8], we predicted that other strains of P[8] can recognize the Lewis-containing LNDFH I in the same ββ site as the BM13851 strain of P[8].

## Discussion

Recently, the importance of HBGAs as cell attachment factors or receptors has been demonstrated for multiple human RV genotypes, however, the molecular level understanding of the glycan binding interactions of the dominant human P[8] RVs to HBGAs remains incompletely understood and under continued investigation. Huang *et al.* originally reported that the P[8] genotype used Le[b] and H type 1 HBGAs as ligands for RV attachment [20], which have been observed by others [46]. However, other groups contradicted these results, reporting that human P[8] RVs do not recognize Le[b] and H type 1 antigens [47]. Different P[8] strains with different affinities to specific HBGAs may account for the contradictory results. Recently, the structural basis of human rotavirus P[8] in recognizing the H type 1 antigen (Fuc-α1,2-Gal-β1,3-GlcNAc) and its precursor lacto-N-biose (Gal-β1,3-GlcNAc) has been characterized by Gozalbo-Rovira et al [38], and lacto-N-fucopentaose 1 (Fucα1-2Galβ1-3GlcNAcβ1-3Galβ1-4Glc) by Sun et al [39]. Both groups demonstrated that different P[8] strains have different binding affinity to H type 1 antigen [38, 39], indicating that the subtle amino acid changes within and outside the defined binding pocket between different P[8] VP8\* strains may explain their different affinities to HBGAs. In an attempt resolve the contradictory results concerning P[8] VP8\* recognition of Le[b] antigens, we used STD-based NMR experiments and HSQC-based NMR titration experiments to study the RV VP8\*-ligand interactions. Our results confirmed that human P[8] VP8\* can bind both Le[b] and H type 1 HBGAs. Crystallization of P[8] with the Le[b] and H type 1 HBGAs are still underway in our laboratory. As an alternative to X-ray crystallography, we used NMR-driven HADDOCK docking to generate data-driven models of the complexes of the P[6] and P[8] VP8\* domains with their respective glycan ligands, which is widely used to characterize protein-ligand interactions when crystals of the complex of interest are unavailable, which is commonly the case for weak binding systems [40–42].

Given that the multiple genotypes in P[II] genogroup exhibit variable host ranges among animal species and different human populations, further clarification of the molecular basis for ligand-controlled host ranges of the major human RVs is important for development of improved vaccines in the future. Structural-based sequence alignment of P[19], P[6], P[4] and

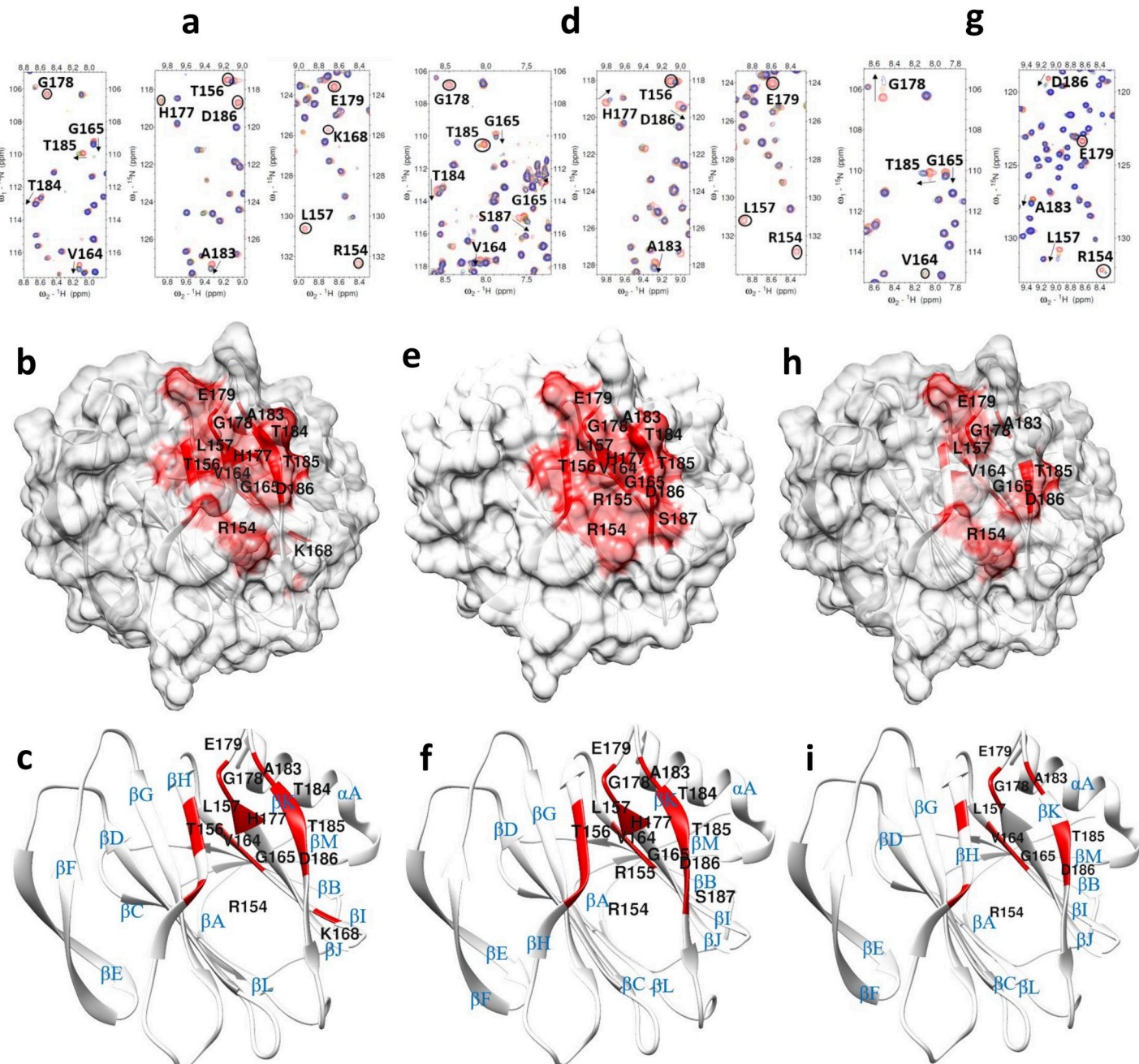

**Fig 6. NMR titration analysis of P[8] VP8\* mutants with LNDFH I glycan.** (a) Chemical shift changes in T78A P[8] VP8* upon addition of LNDFH I. The NMR data correspond to increasing ligand/protein ratios of 0:1 (red), 8:1 (orange), 12:1 (green), and 16:1 (blue). (b) Location of the large chemical shift changes on the T78A P[8] VP8* surface upon binding to LNDFH I. (c) Ribbon Diagram shows the location of large chemical shift changes on the T78A P[8] VP8* surface upon binding to LNDFH I. (d) Chemical shift changes in D79A P[8] VP8* upon addition of LNDFH I. The NMR data correspond to increasing ligand/protein ratios of 0:1 (red), 8:1 (orange), 12:1 (green), and 16:1 (blue). (e) Location of the large chemical shift changes on the D79A P[8] VP8* surface upon binding to LNDFH I. (f) Ribbon Diagram shows the location of large chemical shift changes on the D79A P[8] VP8* surface upon binding to LNDFH I. (g) Chemical shift changes in K168A P[8] VP8* [PDB ID: 2DWR] mutant upon addition of LNDFHI. The NMR data correspond to increasing ligand/protein ratios of 0:1 (red), 8:1 (orange), 12:1 (green), and 16:1 (blue). (h) Location of the large chemical shift changes on the K168A P[8] VP8* surface upon binding to LNDFHI. (i) Ribbon Diagram shows the location of large chemical shift changes on the K168A P[8] VP8* surface upon binding to LNDFHI. The secondary structure is labeled. The threshold values obtained from WT-P[8] titrated with LNDFHI were used to determine the amino acids that show large chemical shift changes, with peaks changes greater than 1.5σ or disappeared are colored with red. The arrows in the spectra show residues that have chemical shifts changes upon titration, and peaks that disappeared upon titration are circled.

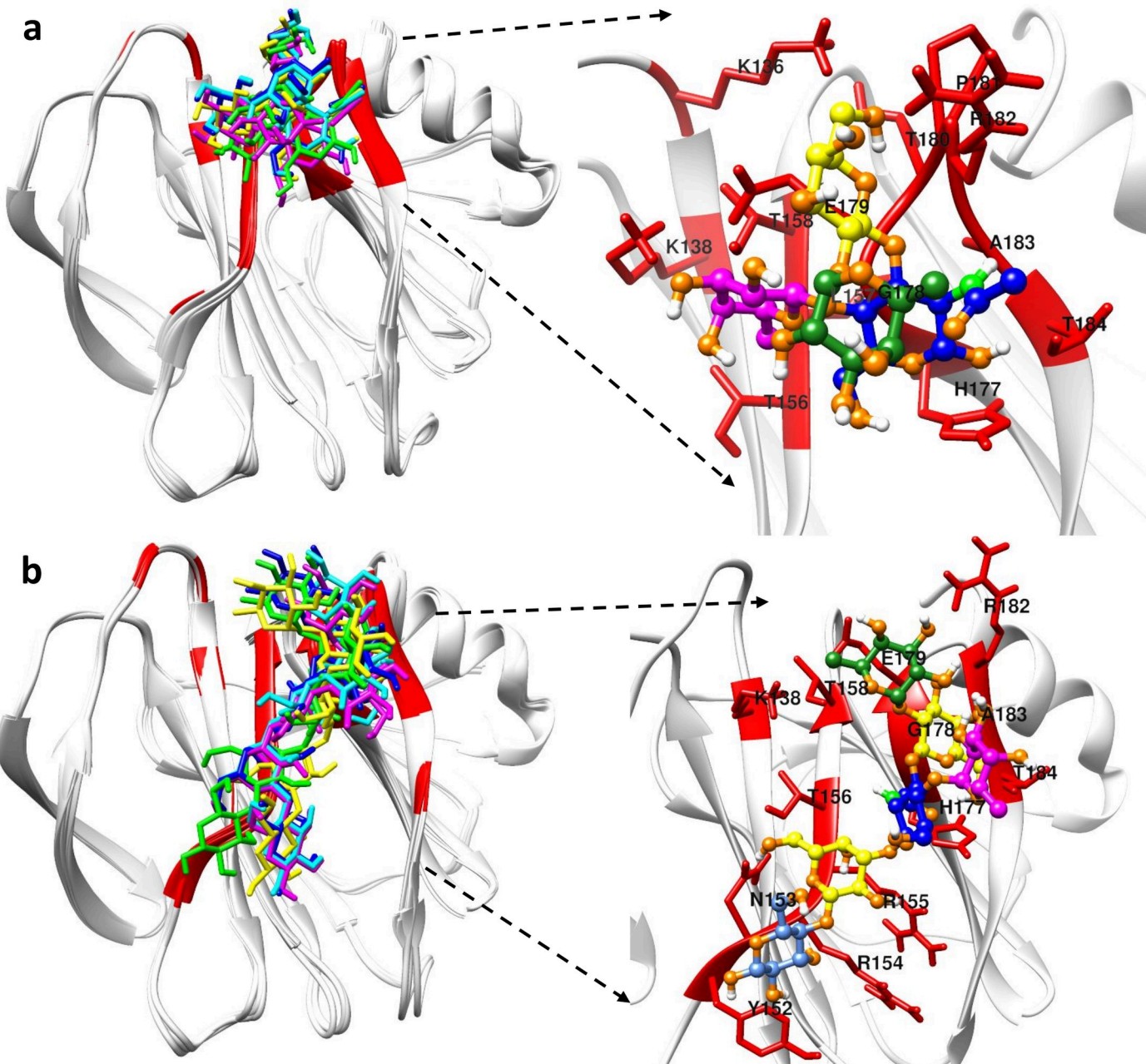

**Fig 7. HADDOCK docking results of P[8] with its glycans.** (a) Cartoon figures in the left panel show the superposition of the top five best-scoring Le$^b$ tetra-saccharide bound P[8] VP8* structures. The right panel in (a) highlights the binding pocket of P[8] in recognizing Le$^b$ tetra-saccharide. (b) The superposition of the top five best-scoring LNDFH I bound P[8] VP8* structures is shown in the left panel. The right panel in (b) highlights the binding pocket of P[8] in recognizing LNDFH I. Red colors in the proteins represent the binding interface. Different colors were used to represent different carbohydrate moiety of the ligand: Lewis fucose (magenta), secretor fucose (green), galactose (yellow), N-acetylglucosamine (blue), glucose (cornflower blue).

P[8] showed that they have high levels of amino acid conservation. From the recent X-ray crys-tallography studies, it was found that P[19], P[6], P[4], P[8] employed the βα binding site to accommodate H type 1 ligand LNFP I [35, 36, 39], which is consistent with our NMR-driven HADDOCK results. Previous homology modeling results suggested that the addition of Lewis fucose via α1,4-linkage to the O4 atom of the GlcNAc residue could cause a binding clash if P

**Table 1. Diffraction data collection and refinement statistics for P[6] BM11596 VP8\* and P[6] BM11596 in complex with LNFP I.**

| | P[6] BM11596 VP8* | P[6] BM11596 VP8*/LNFP I |
|---|---|---|
| **PDB ID** | 6NIW | 6OAI |
| **Crystal parameters** | | |
| Space group | P 1 21 1 | P 1 21 1 |
| Unit cell parameters | | |
| a; b; c (Å) | 56.69, 76.00, 73.88 | 56.69, 75.87, 78.93 |
| α; β; γ (°) | 90.00, 91.89, 90.00 | 90.00, 91.86, 90.00 |
| **Data Collection** | | |
| Wavelength (Å) | 0.97931 | 0.97931 |
| $R_{merge}$ | 0.039 (0.461)[a] | 0.065 (0.316) |
| Resolution (Å) | 73.84–1.55 (1.63–1.55) | 74.89–1.90 (2.00–1.90) |
| Unique reflections | 87179 (12671) | 48808 (7016) |
| Mean $[(I)/\sigma(I)]$ | 13.3 (2.4) | 12.4 (3.0) |
| Completeness | 95.8 (95.4) | 97.3 (96.4) |
| Multiplicity | 3.4 (3.4) | 3.4 (3.2) |
| **Refinement** | | |
| Resolution (Å) | 73.84–1.55 (1.59–1.55) | 74.89–1.90 (2.00–1.90) |
| R-work | 0.199 | 0.197 |
| R-Free | 0.233 | 0.249 |
| Number of protein atoms | 5186 | 5188 |
| Number of amino acid residues | 633 | 633 |
| number of Ligands | 2 (PEG) | 1 (LNFP I) |
| number of Water molecules | 157 | 148 |
| Mean B-values | | |
| Protein | 25.30 | 21.57 |
| Ligands | 37.39 | 34.36 |
| Solvent | 27.68 | 18.71 |
| RMS deviation | | |
| Bond lengths (Å) | 0.014 | 0.014 |
| Bond angles (°) | 1.546 | 1.563 |
| Ramachandran statistics (%) | | |
| Preferred regions | 96.20 | 95.84 |
| Allowed regions | 3.80 | 4.16 |
| Outliers | 0 | 0 |

[a] Values in parentheses are for highest-resolution shell.

[8] used the same binding interface as P[4]/P[6]/P[19] [36]. Our structural studies presented here resolve this perceived conflict, showing that P[8] accommodates binding to the Le^b^-containing ligands at another binding site, namely the ββ binding site that is likely shared with P [4], based on our STD NMR results (S11 Fig). Thus, while P[6] and P[19] use the βα binding site to recognize Lewis negative oligosaccharides, P[8] and P[4] use both the ββ and βα binding sites to recognize Le^b^ tetra-saccharide and LNDFH I and the Le^b^ negative H type 1 saccharides, respectively, which appears to reflect an evolutionary adaptation of RVs driven by selective pressure to accommodate the newly added Lewis epitope. On the other hand, the Lewis epitope may have emerged as a result of co-evolution of humans under selection pressure against P[6]/P[19] RVs and maybe other pathogens that recognized the Lewis negative type 1 HBGAs

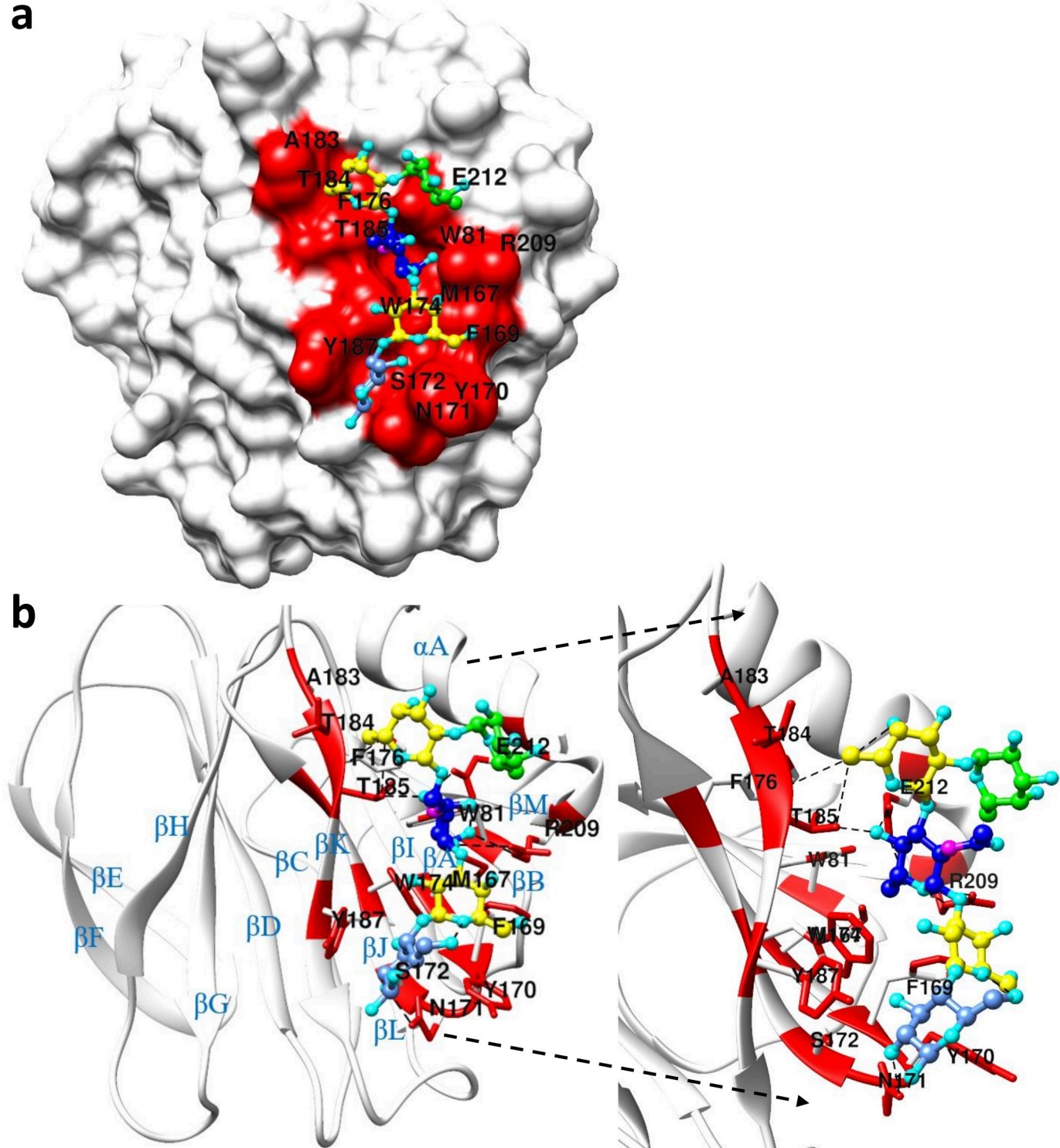

**Fig 8. Crystal structure of LNFP I-P[6] complex.** (a) The surface model of LNFP I-P[6] complex [PDB: 6OAI]. (b): Ribbon diagram shows the LNFP I-P[6] VP8* complex. The right panel in (b) highlights the binding pocket of P[6] VP8* in recognizing LNFP I. Red represents the binding interface. Different colors were used to represent different carbohydrate moiety of the ligand: Green colors the secretor fucose residue, yellow colors the Galactose residue, blue colors the N-Acetylglucosamine residue, cornflower blue colors the Glucose residue.

**Fig 9. Structure-based sequence alignment of VP8* domains.** The VP8* sequences of RVs with known binding sites are included. The VP8* sequences of P[19], P[6], P[4] and P[8] in P[II] genogroup, P[3], P[7] in P[I] genogroup, P[14] in P[III] genogroup and P[11] in P[IV] genogroup are included and the amino acids corresponding to the binding site are highlighted in red. Yellow highlights the potential binding site of Wa and other strains of P[8] VP8* in recognizing HBGAs.

and are lethal to young-age hosts. Structural studies to test these hypotheses are currently underway in our laboratories.

Our data also supports our hypothesis that evolution of P[II] RVs occurs under strong selection by type 1 HBGAs. While most P[I] RVs infect animals, the majority of P[II] RVs infect humans. P[II] RVs were hypothesized to have originated from a P[I] RV with a possible animal host origin and then gained the ability to infect humans by adapting to the polymorphic HBGAs in humans [48]. P[19] RVs are commonly found to infect animals (porcine) but

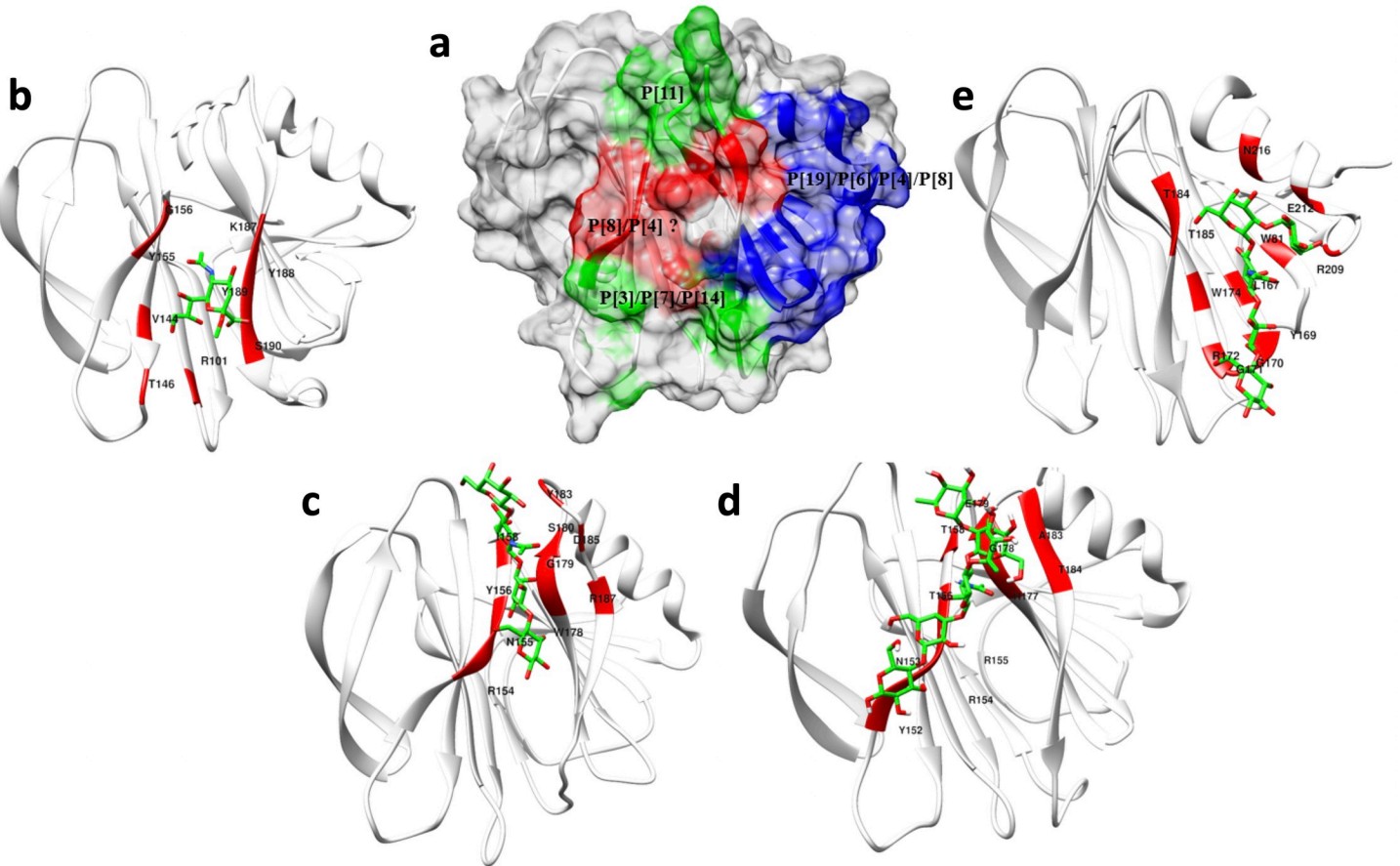

**Fig 10. The binding interface of different genotypes of VP8* in recognizing their corresponding glycans.** (a) Surface-rendered structure of the apo P[6] VP8* domain [PDB ID: 6NIW] as a representative of the VP8* structures depicting a summary of the distinct glycan binding sites identified to date. The red-colored region indicates the ββ binding motif that P[3]/P[7] use in binding sialic acid, P[14] in binding type A HBGAs, P[11] in binding type 1 HBGAs, and P[8] in recognizing Lewis type HBGAs. The blue-colored region indicates the βα binding pocket that the P[19], P[6] and P[4] RVs use in recognizing H-type 1 HBGAs. (b) Detail of the sialic acid binding interface (red) of P[3] [PDB ID:1KQR]. (c) Detail of the LNT binding site of P[11] [PDB ID: 4YFZ]. (d) Detail of the LNDFH I binding interface of P[8]. (e) Detail of the LNFP I binding site of P[4] [PDB ID: 5VX5].

rarely humans. P[19] RVs exhibit binding specificities to mucin core 2 and type 1 HBGA precursors, indicating that P[19] genotype may be positioned at an early evolutionary stage starting from a common P[I] ancestor. Recently, the apo crystal structure of porcine P[6] strain (z84) and human P[6] strain (RV3) have been reported, both demonstrating a galectin-like fold as reported for P[19] [36, 49]. P[6] RVs are commonly found to infect animals (porcine), neonates and young infants [50, 51]. Expression and modification of the precursor moieties in the neonate gut is developmentally regulated, and unbranched type 1 precursor glycans are the most abundant type in the neonate gut [31, 52]. It was found from our NMR titration experiments that P[6] had a much higher affinity to the tetra-saccharide LNT than the penta-saccharide LNFP I, supporting the previous findings that P[6] had a preference to less mature type 1 HBGA precursors. Furthermore, P[6] RVs have an age-specific host range and exhibit cross-species transmission between humans and animals, indicating P[6] may represent an RV evolutionary intermediate that adapted to limited glycan residues shared between porcine and humans. Compared to P[6], the P[4] and P[8] genotypes are genetically closely related and their infections are more commonly detected. The binding affinity of P[8] to the Lewis positive hexa-saccharide LNDFH I ($K_d = 6.3 \pm 0.8$ mM) is much tighter than to the Lewis negative

penta-saccharide LNFP I ($K_d$ = 23.5 ± 7.6 mM) based on HSQC titrations (S12B Fig and S12C Fig), and P[8] accommodated binding to the difucosylated glycans by shifting from the βα to the ββ binding mode (S13 Fig), supporting the view that P[8] and P[4] are more distantly evolved from their P[I] ancestor and have adapted to recognize more mature, evolutionarily evolved, type 1 HBGAs. The fact that P[4] and P[8] are commonly found in humans but rarely in animals would be consistent with their ability and preference to bind additional residues in more complex HBGAs such as is found in the incorporation of Lewis epitopes [26], which are widely distributed in humans but not commonly expressed in animals, such as mice [53].

The elucidation of HBGA-correlated susceptibility to infection with specific RV P-genotypes, mainly associated with secretor status and Lewis phenotype, is important to understand RV disease burden and epidemiology. HBGAs are fucose-containing carbohydrates and act as attachment factors for various pathogenic microorganisms, such as *Helicobacter pylori* [54], Norovirus [55], and Rotavirus. Six types of precursor oligosaccharide chains have been characterized, and type 1 can be found in the epithelium of the respiratory, gastrointestinal, genitourinary tracts and in exocrine secretions [25]. The expression of polymorphic human HBGAs are genetically regulated by specific glycosyltransferase, and the resultant HBGA products for the ABO, H and Lewis families are distributed among the world's population with their frequencies varied from one ethnicity to another [56]. For example, the addition of α-1,2 fucose is catalyzed by the α-1,2 fucosyltransferase that encoded by the FUT2 gene, which is the major determinant of secretor antigens. H-positive phenotypes who contain secretor antigens occurs in about 80% of European and North American populations [57]. The Lewis-positive individuals who contain an active α-1,3 fucosyltransferase (FUT3) enzyme represent about 90% of the general population, but the frequencies are much lower in Africa [58]. The discovery that P[4], P[6], and P[8] human RVs recognize the secretor epitopes of human HBGAs appears to correlate with the predominance of these genotypes in causing the vast majority (>95%) of human infections worldwide.

Epidemiological and biochemical studies suggest that non-secretors and Le[b] negative individuals may be resistant to P[4] and P[8] infections [28, 59, 60]. An epidemiologic survey of children in Tunisia demonstrated that P[8] rotavirus can infect both secretor and non-secretor Lewis antigen-positive individuals [61]. However, other studies demonstrated that gastroenteritis incidence caused by P[8] RVs [28, 62], antibody titers to P[8] genotypes [63], and vaccine take [64] are correlated with the FUT2 phenotype. Our results highlight the importance of the Lewis and secretor status on the P[8] RVs infection as our data demonstrated that P[8] did not recognize the tetra-saccharide LNT without secretor and Lewis epitopes (S12A Fig), but it interacted with the type 1 HBGA with the secretor and Lewis epitopes. Gozalbo-Rovira et al showed that the P[8] VP8* can interact with the type 1 precursor lacto-N-biose (LNB) but with two-fold weaker affinity than H1 trisaccharide [38]. The weaker interaction of P[8] in recognizing type 1 precursor LNB may explain the infection of P[8] rotavirus in non-secretor individuals [61], indicating the relative affinity to different HBGAs may be relevant to the viral susceptibility between secretors and non-secretor individuals. A recent study demonstrated that the susceptibility to RVs was associated with the polymorphism of human HBGAs, however, *in vitro* infection of transformed cell lines was independent of HBGA expression [65]. All the above studies indicated the complex roles of HBGAs in human RVs infection. Non-secretors are also potentially susceptible to infection with prevalent human RVs as other epidemiological studies have demonstrated no significant correlation between secretor status and the susceptibility to P[6] RV infections [27, 28]. These epidemiological studies are supported by our data since we have demonstrated that P[6] recognized the tetra-saccharide LNT that contains the type 1 HBGA sequences without secretor and Lewis epitopes, and the penta-saccharide LNFP I that contains H epitopes but no Lewis epitopes. Compared with human P[4] and P[8] that have a preference to bind to Lewis b antigen, P[6] human RVs have a preference to

bind H type 1 and have a restricted geographic prevalence and are common in the African countries [66, 67]. In these countries, the higher prevalence of P[6] human RVs could be due to the significantly higher rate of Le negative phenotype among the population than in other geographic locations [13, 28].

Even though all of the solved VP8*structures share the same conserved galectin-like fold, they exhibit significant glycan recognition variability by genotype, presumably reflecting evolutionary-dependent differences. While differential immune responses, presence of other co-receptors and variable host factors may all account for the relative distribution of RV genotypes and variable vaccine efficacy in different populations [59], achieving a more complete understanding of the specificity of glycan recognition specificity of VP8*domains may help understand the limited effectiveness of the two current RV vaccines in certain populations and provide clues for the formulation of more effective vaccines. For example, based on relative effectiveness of the two current RV vaccines, Rotarix and RotaTeq, in the developing versus developed countries and their different vaccine designs, a role of the P type (VP4/VP8*) in host immune protection against RVs has been emphasized, which has led to a hypothesis on the lack of cross-protection of the P[8] based Rotarix and RotaTeq against other RV P types that are more commonly seen in the developing countries than in the developed countries could be a major reason of the low effectiveness of both vaccines observed in many developing countries [48], although other factors, such as malnutrition, intestinal microbiota and human maternal milk for children living in many developing countries may also play important roles. Thus, based our emerging understanding of RV diversity, strain-specific host ranges, and RV evolution, an approach that includes other P types in RV vaccines, such as the P[6] and P[11] RVs that are more commonly seen in the developing countries, may improve the global effectiveness of the current vaccines.

## Materials and methods

### Expression and purification of VP8* proteins in *Escherichia coli*

The VP8* core fragments (amino acids 64 to 223) of the human RV P[8] (BM13851) and P[6] (BM11596) with an N-terminal glutathione S-transferase (GST) tag was overexpressed in *Escherichia coli* BL21 (DE3) cells as previously described [20]. The P[8] T78A, D79A, K168A, R154A, H177A, and D186A mutations were generated through overlap-extension site mutagenesis, and the sequences were confirmed at the Center for Bioinformatics and Functional Genomics (CBFG) at Miami University. Cells were grown in 1L Luria broth (LB) medium supplemented with 100 μg mL$^{-1}$ ampicillin at 310 K. When the OD600 reached 0.8, 0.5 mM isopropyl-β-D-thiogalactopyranoside was added to the medium to induce protein expression. The cell pellet was harvested within 12 h after induction and re-suspended in the phosphate-buffered saline (PBS) buffer (140 mM NaCl, 2.7 mM KCl, 10 mM $Na_2HPO_4$, 1.8 mM $KH_2PO_4$, pH 7.3). The cells were lysed by French press (Thermo Fisher Scientific, Waltham, MA), and the supernatant of the bacterial lysate was loaded to a disposable column (Qiagen, Hilden, German) pre-packed with glutathione agarose (Thermo Fisher Scientific). The GST fusion protein of interest was eluted with elution buffer (10 mM reduced glutathione, 50 mM Tris-HCl, pH 8.0). The GST tag of the VP8* protein was removed using the thrombin (Thermo Fisher Scientific) after dialysis into the buffer (20 mM Tris-HCl, 50 mM NaCl, pH 8.0). The VP8* protein without GST tag was further purified by passing the mixture through size exclusion chromatography using a Superdex 200 Hiload (GE Life Science) column. The purified protein was concentrated with an Amicon Ultra-10 (Millipore, Billerica, MA) for future NMR study. The $^{15}$N or $^{15}$N, $^{13}$C-Labled P[6] P[8], and P[8] proteins was made using the $^{15}$N or $^{15}$N,$^{13}$C-Labeled minimal growth medium.

## Inhibition of P[8] RV Wa strain replication in cell culture by synthetic oligosaccharide-BSA or -PAA conjugates in HT29 cells

Trypsin activated Wa viruses were incubated with oligosaccharide conjugate at 37°C for 1 hour and then transferred to DPBS rinsed cell culture wells. After 16 hours all the culture wells were processed by IFA. Serial dilutions of oligosaccharide conjugate at concentration of 250, 125, 62.5, 31.25 and 15.63 µg/mL were tested. About 150 PFU/well Wa inoculum in control wells (without oligosaccharide conjugate) was used.

## Ligand chemical shift assignments

Chemical shift assignments of Le[b] tetra-saccharide, lacto-N-tetraose (LNT), and lacto-N-difucohexaose I (LNDFHI) were completed by collecting and analyzing the following NMR spectra at 20°: 2D $^{1}$H-$^{13}$C heteronuclear single quantum correlation spectroscopy (HSQC), 2D $^{1}$H-$^{13}$C heteronuclear multiple-bond correlation spectroscopy (HMBC), $^{1}$H-$^{1}$H correlation spectroscopy (COSY), $^{1}$H-$^{1}$H total correlation spectroscopy (TOCSY), $^{1}$H-$^{1}$H rotating-frame nuclear Overhauser effect spectroscopy (ROESY), and $^{1}$H-$^{13}$C HSQC-TOCSY. All the glycans were prepared in PBS buffer (pH 7.3). The chemical shifts of lacto-N-fucopentaose I (LNFPI) were previously assigned [26].

## Saturation transfer difference (STD) NMR experiments

All STD NMR spectra were acquired in Shigemi Tubes (Shigemi, USA) on 600-MHz Bruker Avance III and 850-MHz Bruker Avance II NMR spectrometers equipped with conventional 5-mm HCN probes at 283 K with pulse sequence STDDIFFESGP.3. The NMR samples for P[8] VP8* were prepared as 46 µM P[8] VP8* with 2.3 mM LNDFHI (1:50 protein: ligand ratio), and 33 µM P[8] VP8* with 1.86 mM Le[b] tetra-saccharide (1:56 protein: ligand ratio). The NMR samples for P[6] VP* were prepared as 46 µM P[6] with 4.6 mM LNT (1:100 protein: ligand ratio), and 46 µM P[6] with 2.3 mM LNT (1:50 protein: ligand ratio). A control sample was prepared as 40 µM GST with 2.0 mM LNDFHI (1:50 protein: ligand ratio). The STD NMR spectrum of glutathione S-transferase (GST) mixed with LNDFH I was collected as a negative control and it did not show significant signal intensities as expected, validating the specificity of our results. The protein resonances were saturated with a cascade of Gaussian-shaped pulses with a duration of 50 ms, with a total saturation time of 2 s and 4 s. The on resonance was chosen to selectively saturate only the proteins for each sample. The off-resonance saturation was applied to 50 ppm, and a total of 256 scans were acquired. A spin-lock filter with 20 ms duration was applied to suppress the broad protein resonance signals, and residual water signal was suppressed using excitation sculpting with gradients. The STD amplification factors were determined by the equation $(I_{STD}/I_{O}) \times$ molar ratio, where $I_{0}$ are the intensities of the signals in the reference spectrum, and $I_{STD}$ are intensities that obtained by subtracting the on-resonance spectrum from the off-resonance spectrum. The normalized STD amplification percentage for epitope mapping was determined by dividing by the largest STD amplification factor for each ligand.

## Protein backbone chemical shift assignment

The ~0.6 mM double-labeled $^{15}$N, $^{13}$C-P[6] and P[8] VP8* sample were put into 5-mm Shigemi NMR tubes and spectra were collected at 298 K on 600-MHz Bruker Avance III and 850-MHz Bruker Avance II NMR spectrometers equipped with conventional 5-mm HCN probes. Backbone assignments were made based on the following three-dimensional spectra: HNCACB, CBCA(CO)NH, HNCO, HN(CA)CO, HNCA, and HN(CO)CA. Spectra were

processed with the software NMRPipe [68] and visualized with the software SPARKY [69] (https://www.cgl.ucsf.edu/home/sparky/). The backbone $^1$H, $^{13}$C, and $^{15}$N resonance assignments were first made using the PINE server [70], then manually confirmed through SPARKY.

## NMR titration experiments

Chemical shift perturbations were followed by 2D $^1$H-$^{15}$N HSQC for P[8] VP8*, P[8] mutants, and P[6] VP8* upon titration of each glycan ligand. NMR data were collected with samples in PBS buffer in 5-mm Shigemi NMR tubes on 600-MHz Bruker Avance III or 850-MHz Bruker Avance II NMR spectrometers equipped with conventional 5-mm HCN probes. Samples of 0.20 mM $^{15}$N-labeled P[8] VP8* was titrated with LNDFH I at 0, 0.020, 0.040, 0.80, 1.00, 1.20, 1.40, 1.60, 2.00, 2.80, 3.60, 4.40, 5.00 mM. The same ratio of LNDFH I was titrated to each P[8] mutant as the titration of LNDFH I to wild type P[8]. Samples of 0.31 mM $^{15}$N-labeled P[8] VP8* was titrated with Le$^b$ tetra-saccharide at 0, 0.031, 0.15, 0.31, 0.62, 1.24, 2.49, 4.35, 5.60, 6.84, 7.78, 8.71, 9.33 mM. Samples of 0.24 mM $^{15}$N-labeled P[6] VP8* was titrated with LNFP I at 0, 0.024, 0.12, 0.24, 0.48, 0.96, 1.93, 3.37, 4.34, 5.30, 6.03, 6.75, 7.23, 8.44 mM. Samples of 0.33 mM $^{15}$N-labeled P[6] VP8* was titrated with LNT at 0, 0.033, 0.17, 0.33, 0.69, 1.34, 2.67, 4.68, 6.01, 7.35, 8.35 mM. Spectra were process with NMRPipe software [68] and analyzed with SPARKY [69]. The chemical shift changes upon titration was determined by the following formula [71]: $\Delta\delta_{obs} = \sqrt{\frac{1}{2}\left[\delta_H^2 + \left(\alpha \cdot \delta_N^2\right)\right]}$, $\delta_H = \delta_{H\,free} - \delta_{H\,bound}$, $\delta_N = \delta_{N\,free} - \delta_{N\,bound}$. Where $\delta_{H\,free}$ and $\delta_{H\,bound}$ are the backbone amide hydrogen chemical shifts in the absence and presence of ligand, $\delta_{N\,free}$ and $\delta_{N\,bound}$ are the amide nitrogen chemical shifts in the free and bound state. The value of Euclidean weighting correction factor α was set to 0.14 [72]. Standard deviation σ of the Euclidean chemical shift change was calculated and threshold value was chosen from 1.5σ to 2.5σ for the VP8*-HBGA by balancing the specificity and sensitivity. Amino acids with chemical shifts greater than the threshold value were used to calculate the dissociation constants. Dissociation constants were obtained by fitting the following equation: $\Delta\delta_{obs} = \Delta\delta_{max}\left\{([P]_t + [L]_t + K_d) - [([P]_t + [L]_t + K_d)^2 - 4[P]_t[L]_t]^{\frac{1}{2}}\right\}/2P_t$. Where $[P]_t$ and $[L]_t$ represent the total concentrations of protein and ligand, $\Delta\delta_{obs}$ is the change in the observed chemical shift from the free state, $\Delta\delta_{max}$ is the maximum chemical shift change on saturation, $K_d$ represents the dissociation constant.

## Homology modeling and HADDOCK docking

A homology model of the P[8] (BM13851) VP8* protein was built based on the X-ray crystal structure of the Wa strain of RV (PDB: 2DWR). The model building was finished by SWISS-MODEL automated protein structure homology-modeling server [73]. Docking simulation were calculated with HADDOCK2.2 [42, 43]. Ambiguous interactions restraints were defined for active residues in protein identified to be involved in the interactions using NMR titration, and with high solvent accessibility. Solvent accessibility of the proteins were performed by freeSASA [74], and amino acids with solvent accessibility of both main chain and side chain less than average values are filtered out. STD data were used to refine the ambiguous interaction restraints for ligands, and trNOESY distance estimates were used to constrain the conformation of the bound ligand (S1 Table). trNOESY were performed at 1:20 protein/ligand ratio and at a mixing time of 100ms, 150ms, and 200 ms. Longitudinal cross-relaxation rates were obtained by averaging the normalized volume at the three mixing times. From them, intramolecular ligand proton-proton distances were obtained by using the isolated spin pair approximation, and taking the distance of GlcNAc H1-H2 as reference. Each docking calculation

generated 1000/200/200 models for the rigid body docking, semi flexible simulated annealing, and explicit solvent refinement. To optimize protein-ligand docking, the RMSD cutoff for clustering was set to 2.0, the Evdw 1 and Eelec 3 were set to 1.0 and 0.1 respectively, and the initial temperatures for second and third TAD cooling step were set to 500 K and 300 K respectively. The top five models with the lowest/best HADDOCK scores (a linear combination of various energies and buried surface area (S2 Table)) were picked for visualization.

### Protein crystallization

The hanging-drop vapor-diffusion method was used for crystallizing human RV P[6] VP8* protein and co-crystallizing P[6] VP8* in complex with LNFP I (10:1 protein/ligand ratio). Crystals were obtained from drops where 1 μL purified P[6] VP8* was mixed with 1 μL of the reservoir buffer: 0.1 M HEPES sodium pH 7.5, 1.5 M lithium sulfate monohydrate. The crystals were harvested, cryo-protected using mother liquor supplemented with 10% (v/v) glycerol and immediately flash-cooled in liquid nitrogen.

### Data collection, processing, and structure determination

Diffraction data were collected at the Advanced Photon Source (APS) beamline 31-ID-D, Argonne National Laboratory, in Chicago, Illinois. A total of 180 images were collected using 0.2˚ oscillation with 0.24 s exposures per image. The images were integrated with MOSFLM [75], and scaled with SCALA [76]. Molecular replacement was performed with PHASER [77] using the coordinates of chain A from 5VX8 [36] as the search model. Iterative model building was manually carried out in COOT [78], and refinements using 5% of reflection in Free-R set were carried out in REFMAC [79] implemented in the CCP4 suite [80]. The structure quality was assessed using MolProbity [81]. Final model and scaled reflection data were deposited at the Protein Data Bank (PDB ID: 6NIW, 6OAI). The visualization and investigation of the final model was analyzed using Chimera [82] and LigPlot+ [83]. Sequence alignment was finished by Clustal Omega [84].

### Supporting information

**S1 Fig. NMR spectra of glycans.** Two-dimensional heteronuclear single quantum coherence spectroscopy of (a) Le$^b$ tetra-saccharide. (b) LNDFH I. (c) LNT. (d) LNFP I. Individual resonance assignments are indicated by labels. Red colors indicate positive signals, green colors indicate negative signals.
(TIF)

**S2 Fig. NMR spectra of VP8* protein.** The $^1$H-$^{15}$N HSQC spectrum of (a) P[8] VP8* and (b) P[6] VP8*. Individual resonance assignments are indicated by labels.
(TIF)

**S3 Fig. Chemical shift change profile of P[8] VP8* upon addition of glycans.** (a) Euclidean chemical shift changes of amino acids in P[8] VP8* upon addition of Le$^b$ tetra-saccharide. Blue line represents the standard deviation 1.5σ of Euclidean chemical shift change, and green and red lines represent 2σ and 2.5σ threshold values respectively. (b) Euclidean chemical shift changes of amino acids in P[8] VP8* upon addition of LNDFH I. Blue line represents the standard deviation σ of Euclidean chemical shift change, and green and red lines represent 1.5σ and 2σ threshold values, respectively.
(TIF)

**S4 Fig. Dissociation constants from NMR titration of P[8].** Global fitting for (a) P[8] and Le$^b$ tetra-saccharide, samples of 0.31 mM $^{15}$N-labeled P[8] VP8* was titrated with Le$^b$ tetra-saccharide at 0, 0.031, 0.15, 0.31, 0.62, 1.24, 2.49, 4.35, 5.60, 6.84, 7.78, 8.71, 9.33 mM and (b) P[8] and LNDFH I, Samples of 0.20 mM 15N-labeled P[8] VP8* was titrated with LNDFH I at 0, 0.020, 0.040, 0.80, 1.00, 1.20, 1.40, 1.60, 2.00, 2.80, 3.60, 4.40, 5.00 mM.
(TIF)

**S5 Fig. Chemical shift change profile of P[6] VP8*** **upon addition of glycans.** (a) Euclidean chemical shift changes of amino acids in P[6] VP8* upon addition of LNT. (b) Euclidean chemical shift changes of amino acids in P[6] VP8* upon addition of LNFP I. Blue line represents the standard deviation σ of Euclidean chemical shift change, and green and red lines represent 1.5σ and 2σ threshold values, respectively.
(TIF)

**S6 Fig. Dissociation constants from NMR titration of P[6].** Global fitting to extract the dissociation constants from NMR titration for (a) P[6] with LNT. Samples of 0.33 mM 15N-labeled P[6] VP8* was titrated with LNT at 0, 0.033, 0.17, 0.33, 0.69, 1.34, 2.67, 4.68, 6.01, 7.35, 8.35 mM. and (b) P[6] with LNFP I. Samples of 0.24 mM 15N-labeled P[6] VP8* was titrated with LNFP I at 0, 0.024, 0.12, 0.24, 0.48, 0.96, 1.93, 3.37, 4.34, 5.30, 6.03, 6.75, 7.23, 8.44 mM.
(TIF)

**S7 Fig. Chemical shift change profile of P[8] mutants upon addition of LNDFH I.** Euclidean chemical shift changes of amino acids in different P[8] VP8* mutants upon addition of LNDFH I for (a) K168A-P[8], (b) D79A-P[8], (c) T78A-P[8], (d) R154A-[8], (e) D186A-P[8], (f) H177A-P[8]. The threshold was obtained from the titration of the same ratio of LNDFH I into wt-P[8], with blue line represents the standard deviation σ of Euclidean chemical shift change, and green and red lines represent 1.5σ and 2σ threshold values, respectively.
(TIF)

**S8 Fig. HADDOCK docking results of P[6] with its glycans.** (a) Cartoon figures show the superposition of the top five best-scoring LNT bound P[6] VP8* structures from HADDOCK docking. (b) The superposition of the top five best-scoring LNFP I bound P[6] VP8* structures from the HADDOCK docking results. Red colors represent the binding interface.
(TIF)

**S9 Fig. The P[6]-LNFP I interaction diagrams generated by LigPlot+.** The diagram shows the specific interaction between P[6] VP8* and LNFP I. All the amino acid residues and saccharide moieties involved in the interactions are labeled. Hydrogen bond interactions are shown as green dashed lines between the respective donor and acceptor atoms along with the bond distance. The van der Walls contacts are indicated by an arc with spokes radiating towards the ligand atoms they contact.
(TIF)

**S10 Fig. STD NMR spectroscopy for the control experiments.** The bottom spectrum in (a) is the $^1$H NMR reference spectrum of glutathione S-transferase (40 μM) with LNDFH I (2.0 mM). The upper spectrum in (a) is the STD NMR spectrum of GST (40 μM) with LNDFHI (2.0 mM). The protein was saturated with a cascade of 40 Gaussian-shaped pulses at 6.8 ppm., and The off-resonance was set to 50 ppm.
(TIF)

**S11 Fig. STD NMR spectroscopy for P[4] with its glycans.** (a) NMR spectra pf LNDFH I in complex with P[4] VP8*. The bottom spectrum in (a) is the $^1$H NMR reference spectrum of P

[4] VP8* (50 μM) with LNDFH I (2.5 mM).The middle spectrum in (a) is the STD NMR spectrum of P[4] VP8* (50 μM) with LNT (2.5 mM). The protein was saturated with a cascade of 40 Gaussian-shaped pulses at -0.20 ppm, and the off-resonance was set to 50 ppm. The upper spectrum in (a) is the expansion of the STD NMR from 4.0 to 2.6 ppm. (b) The epitope map of LNDFH I when bound to P[4] based on the STD effects: red, strong STD NMR effects (>50%); blue, medium STD NMR effects (30%-50%); green, weak STD NMR effects (<30%). (c) NMR spectra pf LNFP I in complex with P[4] VP8*. The bottom spectrum in (c) is the ${}^1$H NMR reference spectrum of P[4] VP8* (50 μM) with LNFP I (2.5 mM). The middle spectrum in (c) is the STD NMR spectrum of P[4] VP8* (50 μM) with LNFPI (2.5 mM). The protein was saturated with a cascade of 40 Gaussian-shaped pulses at -0.20 ppm, and the off-resonance was set to 50 ppm. The upper spectrum is the expansion of the STD NMR from 4.0 to 2.6 ppm. (d) The epitope map of LNFP I when bound to P[4] based on the STD effects: red, strong STD NMR effects (>85%); blue, medium STD NMR effects (40%-85%); green, weak STD NMR effects (<40%). "*" means the overlap of signals in the STD NMR spectrum. "d" indicates a downfield proton, "u" indicates an upfield proton.
(TIF)

**S12 Fig. Chemical shift change profile of P[8] VP8* upon addition of glycans.** (a) Euclidean chemical shift changes of amino acids in P[8] VP8* upon addition of LNT. (b) Euclidean chemical shift changes of amino acids in P[8] VP8* upon addition of LNFP I. Blue line represents the standard deviation σ of Euclidean chemical shift change, and green and red lines represent 1.5σ and 2σ threshold values respectively. Black line represents the 1.5σ of Euclidean chemical shift change of P[8] upon addition of LNDFH I. (c) global fitting for P[8] and LNFP I, to get the dissociation constant. Samples of 0.20 mM 15N-labeled P[8] VP8* was titrated with LNFP I at 0, 0.020, 0.040, 0.80, 1.00, 1.20, 1.40, 1.60, 2.00, 2.80, 3.60, 4.40, 5.00 mM.
(TIF)

**S13 Fig. Titration results of P[8] VP8* with its glycans.** (a) Chemical shift changes in P[8] VP8* [PDB ID: 2DWR] upon addition of LNFP I. Titrations were followed by acquisition of two-dimensional ${}^1$H-${}^{15}$N HSQC spectra of mM ${}^{15}$N-labeled P[8] VP8*. The NMR data correspond to increasing ligand/protein ratios of 0:1 (red), 8:1 (orange), 18:1 (green), and 25:1 (blue). (b) Location of the large chemical shift changes on the P[8] VP8* surface [PDB ID: 2DWR] upon binding to LNFP I. Amino acids with chemical shift changes greater than 2σ or disappeared after titration are colored with red. (c) Ribbon Diagram shows the location of large chemical shift changes on the P[8] VP8* surface upon binding to LNFP I. The secondary structure is labeled.
(TIF)

**S1 Table. Distance values obtained from the trNOESY for each complex pair.**
(XLSX)

**S2 Table. Summary of the HADDOCK score values, Van Der Waals Energy, electrostatic energy, and buried surface area of the top five structures from the HADDOCK docking for each complex pair.**
(XLSX)

## Acknowledgments

This research used resources of the Advanced Photon Source, a U.S. Department of Energy (DOE) Office of Science User Facility operated for the DOE Office of Science by Argonne National Laboratory under Contract No. DE-AC02-06CH11357. Use of the Lilly Research

Laboratories Collaborative Access Team (LRL-CAT) beamline at Sector 31 of the Advanced Photon Source was provided by Eli Lilly Company, which operates the facility. We acknowledge the assistance of Dr. Mueller with the use of Miami University's Redhawk HPC cluster for conducting the HADDOCK simulations.

## Author Contributions

**Conceptualization:** Shenyuan Xu, Xi Jiang, Michael A. Kennedy.

**Data curation:** Shenyuan Xu.

**Formal analysis:** Shenyuan Xu, Xi Jiang, Michael A. Kennedy.

**Funding acquisition:** Xi Jiang, Michael A. Kennedy.

**Investigation:** Shenyuan Xu, Luay U. Ahmed, Michael Robert Stuckert, Kristen Rose McGinnis, Yang Liu, Ming Tan, Pengwei Huang, Weiming Zhong, Dandan Zhao, Xi Jiang, Michael A. Kennedy.

**Methodology:** Shenyuan Xu, Xi Jiang, Michael A. Kennedy.

**Project administration:** Xi Jiang, Michael A. Kennedy.

**Resources:** Xi Jiang, Michael A. Kennedy.

**Supervision:** Xi Jiang, Michael A. Kennedy.

**Validation:** Shenyuan Xu, Xi Jiang, Michael A. Kennedy.

**Visualization:** Shenyuan Xu, Michael A. Kennedy.

**Writing – original draft:** Shenyuan Xu, Xi Jiang, Michael A. Kennedy.

**Writing – review & editing:** Shenyuan Xu, Xi Jiang, Michael A. Kennedy.

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
