## [Decision Letter · Decision Letter 0]

29 Dec 2019

Dear Dr. Kennedy,

Thank you very much for submitting your manuscript "Molecular basis of P[6] and P[8] major human rotavirus VP8* domain recognition of histo-blood group antigens" (PPATHOGENS-D-19-02135) for review by PLOS Pathogens. Your manuscript was fully evaluated at the editorial level and by independent peer reviewers. The reviewers appreciated the attention to an important problem, but raised some substantial concerns about the manuscript as it currently stands. These issues must be addressed before we would be willing to consider a revised version of your study. We cannot, of course, promise publication at that time.

We therefore ask you to modify the manuscript according to the review recommendations before we can consider your manuscript for acceptance. Your revisions should address the specific points made by each reviewer.

(1) A letter containing a detailed list of your responses to the review comments and a description of the changes you have made in the manuscript. Please note while forming your response, if your article is accepted, you may have the opportunity to make the peer review history publicly available. The record will include editor decision letters (with reviews) and your responses to reviewer comments. If eligible, we will contact you to opt in or out.

(2) Two versions of the manuscript: one with either highlights or tracked changes denoting where the text has been changed; the other a clean version (uploaded as the manuscript file).

Additionally, to enhance the reproducibility of your results, PLOS recommends that you deposit your laboratory protocols in protocols.io, where a protocol can be assigned its own identifier (DOI) such that it can be cited independently in the future. For instructions see http://journals.plos.org/plospathogens/s/submission-guidelines#loc-materials-and-methods

We hope to receive your revised manuscript within 60 days. If you anticipate any delay in its return, we ask that you let us know the expected resubmission date by replying to this email. Revised manuscripts received beyond 60 days may require evaluation and peer review similar to that applied to newly submitted manuscripts.

[LINK]

Sincerely,

John T. Patton, PhD

Associate Editor

PLOS Pathogens

David Wang

Section Editor

PLOS Pathogens

Kasturi Haldar

Editor-in-Chief

PLOS Pathogens

orcid.org/0000-0001-5065-158X

Grant McFadden

Editor-in-Chief

PLOS Pathogens

orcid.org/0000-0002-2556-3526

I am returning your manuscript along with the reviewers’ comments and suggestions. All three reviewers indicate that the manuscript provides significant insight into interactions of the rotavirus P[6] and P[8] VP8* subunit with cellular glycans. However, as detailed in their comments, reviewers 2 and 3 have brought up several major issues with the manuscript that will need to be thoroughly addressed, and the manuscript appropriately modified, before the work can be further considered for publication. Please note that reviewers 2 and 3 have both noted that the authors have failed to cite or discuss their findings in context of those previously reported by others (e.g., Gozalbo-Rovira et al, 2019; etc). This point is particularly important, given that in some cases, the authors' findings and conclusions differ. Please also review for grammatical mistakes and update information included in the manuscript. For example, the number of VP4 genotypes is now greater than 50, and not 40 as indicated in the Introduction. Note that the revised manuscript will be re-evaluated by the more critical reviewers upon resubmission.

Reviewer's Responses to Questions

**Part I - Summary**

Reviewer #1: Xu and colleagues report a structural analysis of the interaction between glycans and the VP8* domain of the spike protein from major human rotavirus strains. The main originality of the study is that it was largely performed by NMR rather than by X-ray crystallography only. Another important novel aspect of this manuscript is that it reports interactions between VP8* of the P[8] type and difucosylated oligosaccharides characteristic of the so-called Lewis b histo-blood group antigen. This aspect is of outmost importance with regard to the genetic component of rotavirus infection that has been described across several epidemiological studies and since P[8] strains are responsible for about 75% of all rotavirus gastroenteritis cases worldwide. Here the authors show that interactions with these difucosylated tetra or hexasaccharides occur through amino-acids located on two beta sheets and do not involve the alpha helix described in two previous studies on P[8] VP8* based on analysis of the interaction with non-fucosylated (type 1 precursor) or monofucosylated (H type 1) oligosaccharides. Validation of the authors conclusions regarding the P[8] VP8* is achieved by comparing the data of the NMR analysis and X-ray analysis of P[6] VP8* that has also previously been reported with similar non-fucosylated or monofucosylated oligosaccharides. Overall the work is sound and the results shed light on a rather complicated topic. Critically, it provides for the first time structural data consistent with the epidemiologic studies that showed the importance of the both the secretor and Lewis phenotypes in P[8] rotavirus gastroenteritis as initially documented by the study of Nordgren et al in 2014 (cited as reference 28) and confirmed by several additional studies since then.

Reviewer #2: The study nicely demonstrates how crystallography and NMR spectroscopy can complement each other to study binding of glycans to proteins, in this case viral capsid proteins. One feature that makes quantification of glycan-protein interactions more tricky than other biomolecular interactions is the inherent weak affinity. Here, chemical shift perturbation experiments may be considered as a gold standard, providing accurate affinity data. Actually, there is only little resilient data on this particular issue out in the literature, and, therefore, I believe this work will whet the appetite for more studies into this direction.

The results are scientifically sound, and the presentation of data is very clear. The novelty of this study lies in the opportunity to link glycan recognition to the spread of disease in different populations, as the authors discuss.

I really cannot see any weaknesses.

I strongly recommend publication of this nice piece of work.

Reviewer #3: This work by Xu et al. investigated the molecular basis of glycan-binding by human rotavirus P[8] and P[6] VP8*s using STD-NMR, HADDOCK, and crystallography. Using NMR, docking, and mutagenesis experiments, the authors found a novel glycan-binding site on P[8] VP8* that is distinct from the recently reported glycan-binding site on P[4], P[6] and P[8] VP8*s. They also solved the crystal structure of P[6] VP8* in complex with LNFP1.

However, the literature review of the manuscript is insufficient. For instance, it is outdated information that “no structural data regarding the P[8] RV-glycan interactions is currently available”(page 6, paragraph #2).

Please see:

1) Unraveling the role of the secretor antigen in human rotavirus attachment to histo-blood group antigens. Gozalbo-Rovira R, et al. PLoS Pathog 2019. PMID 31226167

2) Structural Basis of Glycan Recognition in Globally Predominant Human P[8] Rotavirus. Sun X, et al. Virol Sin 2019. PMID 31620994

The authors need to discuss/compare the new results with the previously reported P[8]-glycan structures. Differences in P[8] sequences that lead to different glycan-binding specificity/sites may be discussed.

**Part II – Major Issues: Key Experiments Required for Acceptance**

Reviewer #1: - The paper by Gozalbo-Rovira published this year in PLoS Pathogens (https://doi.org/10.1371/journal.ppat.1007865) should be cited and critically discussed since it describes the carbohydrate-binding site of a P[8] VP8* through co-crystallization with the type 1 precursor disaccharide or the H type 1 trisaccharide. It involves both the alpha helix and beta strands, consistent with the conclusions of Hu et al, cited as reference 36. These two papers claim that the P[8] and P[6] binding sites are similar and that the fucose alpha1,4 linked to the GlcNAc residue of the type 1 precursor clashes in that binding site, hampering recognition of Lewis type glycans. This is consistent with the authors’ present data regarding P[6] VP8* as well as the above-mentioned epidemiological study of Nordgren et al and the saliva binding data reported by Barbé et al (Scientific Reports, 2018) also regarding P[6] VP8*. However, it is clearly at odds with the authors’ conclusions and with other previous studies that showed binding of P[8] VP8* to the difucosylated Lewis b as well as to saliva from Lewis b individuals but not to saliva from individuals lacking Lewis b (Huang et al J Virol 2012 ; Barbé et al Scientific Reports 2018). It appears that in absence of the alpha1,4-linked fucose residue, P[8] VP8* interact with type 1 oligosaccharides (based on lacto-N-biose) at the same site as P[6] VP8*, whilst the presence of that Lewis fucose, as shown here, shifts positioning of the oligosaccharide toward the two beta sheets mode. That should have a major impact on the virus attachment to natural glycans and the ensuing of infection and needs being properly discussed.

- The authors confirm here previously reported binding of P[8] VP8* to both Lewis b and H type 1. However, the same VP8* attach to saliva samples from Lewis b expressing individuals only and Lewis negative individuals appear largely spared by P[8] rotavirus. How can this be reconciled since an effect of the secretor status only should be visible if binding to H type 1 and Lewis b are equivalent? Could the STD effect observed with the Fuc-IV fucose residue (Fig. 2) be sufficient to increase affinity? A comparison of the dissociation constant for LNF I with that reported for LNDFH I is necessary to strengthen the manuscript by clarifying this issue.

Reviewer #2: None

Reviewer #3: By using STD-NMR, the authors showed that Leb and LNDFH1 bound to P[8] VP8* mainly use the type-1 precursor motif, while the Lewis and secretor fucose residues are also involved in the interaction. Infectivity experiments using Leb, LNDFH1, and a negative control glycan should be carried out to show the biological relevance of this study.

**Part III – Minor Issues: Editorial and Data Presentation Modifications**

Reviewer #1: - The authors compared X-ray crystallography and NMR analyses for the P[6] VP8*-glycan interaction. Why not perform an X-ray crystallography analysis of the difucosylated oligosaccharides with a P[8] VP8*? If that failed, it would be worth mentioning.

- In the abstract with several additional occurrences in the manuscript, the authors state that HBGAs have been identified as receptors for human RV strains. It would be wiser to speak about attachment factors rather than receptors since these glycans do not seem to be required for in vitro infection.

- Legend to Fig 2 indicates P[6] instead of P[8], please correct.

- On page 15, first paragraph, Fig 9c should be Fig 10.

- On page 15, the last sentences of the results section require language edition.

- On page 17, the difference in dissociation constants between Lewis b tetrasaccharide and LNDFH I is used to argue that P[8] and P[4] strains have adapted to recognize more mature type 1 HBGAs. However, the affinity difference is more likely due to the contribution of the lactose unit in LNDFH I. The Leb tetrasaccharide does not correspond to a naturally occuring molecule. Therefore an affinity difference between the hexasaccharide and tetrasaccharide forms of Leb cannot be ascribed to any particular adaptation. Nonetheless, I agree that an adaptation likely took place to accommodate fucosylated glycans by shifting from the βα to the ββ binding domains.

- References numbering should be checked as there are errors. For example, reference 45 cited in the text as Jiang et al is #46 in the reference list.

- Reference 27 (Ayouni et al) is cited as showing that nonsecretors individuals may be resistant to P[8] and P[4] infections. Unfortunately, that study reported a lack of association between the secretor phenotype and rotavirus infection. It should be correctly cited or replaced by another reference corresponding to a study reporting the correct association between the secretor status and P[8] infection.

- From the colors on Fig 7, only 4 modeled Leb or LNDFH I can be distinguished whilst 5 are mentioned in the text and legend to the figure? Is it simply a difficulty to visualized 5 superimposed structures and can this be improved?

- It would be useful to expand the Lewis b binding area shown on Fig 7c in order to better visualize the positioning of the two fucose residues.

- Adding the meaning of the red and green colors assigned to individual resonance assignments on Fig S1 would help the reader.

Reviewer #2: There are only two minor points that need clarification:

1. The bmrb accession code given on p.9, l. 2 is certainly not correct.

2. Some of the dissociation constants reported are in the two-digit millimolar range. Can the authors comment in some more detail - maybe in the supplementary part - up to which concentration they have done the CSP titrations. Some of the titration curves look rather "linear".

Reviewer #3: The authors referred to the glycan-binding sites as “beta-alpha binding domain” and “beta-beta binding domain”. However, protein domains, such as VP8* domain, form compact 3D structures that are independently stable and folded. Have the authors expressed the “beta-alpha” sheets or “beta-alpha” independently? If not, it is more accurate to refer to them as “sites” instead of “domains”.

The authors cited "Fig1a" or "Fig1c" in the text. However, there are no "a","b"..lables in Figure 1.

The crystallographic statistics table (Page 12-14) is out of place and does not have a table legend.

PLOS authors have the option to publish the peer review history of their article (what does this mean?). If published, this will include your full peer review and any attached files.

Reviewer #1: Yes: Jacques Le Pendu

Reviewer #2: Yes: Thomas H. Peters

Reviewer #3: No

---

## [Editor Report · Decision Letter 1]

5 Feb 2020

Dear Dr. Kennedy,

We are pleased to inform you that your manuscript 'Molecular basis of P[II] major human rotavirus VP8* domain recognition of histo-blood group antigens' has been provisionally accepted for publication in PLOS Pathogens.

Before your manuscript can be formally accepted you will need to complete some formatting changes, which you will receive in a follow up email. A member of our team will be in touch within two working days with a set of requests.

Best regards,

John T. Patton, PhD

Associate Editor

PLOS Pathogens

David Wang

Section Editor

PLOS Pathogens

Kasturi Haldar

Editor-in-Chief

PLOS Pathogens

orcid.org/0000-0001-5065-158X

Michael Malim

Editor-in-Chief

PLOS Pathogens

orcid.org/0000-0002-7699-2064

I appreciate the considerable effort the authors have made in revising the manuscript, including performing new experiments and analyses, and addressing issues brought up by the reviewers. This very interesting manuscript is now acceptable for publication. Thank you for submitting the work to PLoS Pathogen.
---

## [Editor Report · Acceptance letter]

5 Mar 2020

Dear Dr. Kennedy,

We are delighted to inform you that your manuscript, "Molecular basis of P[II] major human rotavirus VP8* domain recognition of histo-blood group antigens," has been formally accepted for publication in PLOS Pathogens.

Best regards,

Kasturi Haldar

Editor-in-Chief

PLOS Pathogens

orcid.org/0000-0001-5065-158X

Michael Malim

Editor-in-Chief

PLOS Pathogens

orcid.org/0000-0002-7699-2064